
# Low flow estimation beyond the mean - expectile loss and extreme gradient boosting for spatio-temporal low flow prediction in Austria

Johannes Laimighofer[1], Michael Melcher[2], and Gregor Laaha[1]

[1]University of Natural Resources and Life Sciences, Vienna, Department of Landscape, Spatial and Infrastructure Sciences, Institute of Statistics, Peter-Jordan-Strasse 82/I, 1190 Vienna, Austria
[2]Institute of Information Management, FH JOANNEUM – University of Applied Sciences, Graz, Austria

**Correspondence:** Johannes Laimighofer (johannes.laimighofer@boku.ac.at)

**Abstract.** Accurate predictions of seasonal low flows are critical for a number of water management tasks that require inferences about water quality and the ecological status of water bodies. This paper proposes an extreme gradient tree boosting model (XGBoost) for predicting monthly low flow in ungauged catchments. Particular emphasis is placed on the lowest values (in the magnitude of annual low flows and below) by implementing the expectile loss function to the XGBoost model. For this purpose, we test expectile loss functions based on decreasing expectiles (from $\tau = 0.5$ to $0.01$) that give increasing weight to lower values. These are compared to common loss functions such as mean and median absolute loss. Model optimization and evaluation is conducted using a nested cross validation approach that includes recursive feature elimination to promote parsimonious models. The methods are tested on a comprehensive dataset of 260 stream gauges in Austria covering a wide range of low flow regimes. Our results demonstrate that the expectile loss function can yield high prediction accuracy, but the performance drops sharply for low expectile models. With a median $R^2$ of 0.67, the $0.5$ expectile yields the best performing model. The $0.3$ and $0.2$ perform slightly worse, but still outperform the common median and mean absolute loss functions. All expectile models include some stations with moderate and poor performance that can be attributed to some systematic error, while the seasonal and annual variability is well covered by the models. Results for the prediction of low extremes show an increasing performance in terms of $R^2$ for smaller expectiles $(0.01, 0.025, 0.05)$, though leading to the disadvantage of classifying too many extremes for each station. We found that the application of different expectiles leads to a trade-off between overall performance, prediction performance for extremes, and misclassification of extreme low flow events. Our results show that the $0.1$ or $0.2$ expectiles perform best with respect to all three criteria. The resulting extreme gradient tree boosting model covers seasonal and annual variability nicely and provides a viable approach for spatio-temporal modelling of a range of hydrological variables representing average conditions and extreme events.

## 1 Introduction

Prediction of low flow in ungauged basins is a basic requirement for many water management tasks (Smakhtin, 2001). Current estimation procedures aim to estimate some long-term average low flow characteristic, such as the low flow quantile Q95 or the 7-day minimum flow, often calculated for a return period of 10 years ($Q_{7,10}$) (Salinas et al., 2013). These signatures are either predicted by physically based-models (Euser et al., 2013) or statistical methods, such as geostatistical methods (e.g.





Castiglioni et al., 2009, 2011; Laaha et al., 2014) and regression-based models (e.g. Laaha and Blöschl, 2006, 2007; Tyralis et al., 2021b; Worland et al., 2018; Ferreira et al., 2021; Laimighofer et al., 2022). Recently, the prediction of seasonal low flow characteristics has gained increasing interest. Knowing the seasonal (e.g. monthly) distribution of low flows is necessary, for example, when assessing the water quality or ecological status of water bodies, as low discharges combined with temperature can yield to a cascade of hydrochemical processes that vary with the season. Such temporal low flow characteristics require a

new class of models that take temporal signals of predictors into account. Assessment of low flow on a temporal scale is mostly based on empirical characteristics of the modeled hydrograph (e.g. Shrestha et al., 2014; Huang et al., 2017; Lees et al., 2021), with some exceptions where the accuracy of observations below a specific threshold are considered (e.g. Onyutha, 2016). All these approaches show a much higher bias for low flows than for high flows. This is often a consequence of a loss function that emphasizes high flows while giving too little weight to the low flow events (Staudinger and Seibert, 2014; Staudinger et al.,

2011). Although there exist several approaches for modelling monthly or annual streamflow records (e.g. Vandewiele and Elias, 1995; Steinschneider et al., 2015; Yang et al., 2017; Ossandón et al., 2022; Pumo et al., 2016; Vicente-Guillén et al., 2012; Cutore et al., 2007; Lima and Lall, 2010; Roksvåg et al., 2020), we noticed a significant research gap in modelling monthly low flow, which to our knowledge has not been investigated to date.

Recently, data-driven models have gained interest for prediction of daily discharge in ungauged basins because of their fast
implementation, need of less data and good prediction performance. These approaches consider a wide range of models, e.g. long short term memories (LSTM, Kratzert et al., 2019a, b; Lees et al., 2021) or artificial neural networks (ANN, Solomatine and Ostfeld, 2008; Dawson and Wilby, 2001; Abrahart et al., 2012). Similar methods are applied to lower temporal resolutions (monthly or annual) of streamflow data, where either parameters of hydrological models are interpolated in space (Yang et al., 2017; Vandewiele and Elias, 1995; Steinschneider et al., 2015), or different statistical methods are applied. Considering

the data-driven models a common approach is to fit independent models to each station (e.g. Shortridge et al., 2016; Parisouj et al., 2020; Chang and Chen, 2018). Such an approach seems not efficient, as spatial correlations of time series at neighboring stations are not considered in parameter estimation. This may lead to spatially inconsistent predictions that also have lower accuracy. Few approaches have been proposed that treat spatio-temporal flow indices in a single, spatio-temporal framework (e.g. Ossandón et al., 2022; Vicente-Guillén et al., 2012; Lima and Lall, 2010; Roksvåg et al., 2020; Pumo et al., 2016; Cutore

et al., 2007). These studies either used a combination of deterministic models with kriging (Vicente-Guillén et al., 2012), time series models (Pumo et al., 2016), or Bayesian hierachical models (Ossandón et al., 2022; Lima and Lall, 2010; Roksvåg et al., 2020). Surprisingly little efforts have been undertaken to use statistical learning models for spatio-temporal flow patterns. One exception is Cutore et al. (2007), which tested an artificial neural network (ANN) model for mean monthly flow and compared it against various regression approaches. They found that a single ANN outperforms methods where regression parameters are

interpolated in space, or single multivariable regression.

In this study we propose to use the extreme gradient boosting model (XGBoost Chen and He, 2015; Chen and Guestrin, 2016) for modeling space-time patterns of monthly low flows. XGBoost is an ensemble of boosted regression trees and a common model in hydrology (Zounemat-Kermani et al., 2021). Applications range from e.g. modeling water quality (e.g. Lu and Ma, 2020) to estimate groundwater salinity (Sahour et al., 2020). Additionally, XGBoost has shown to be a suitable model for





streamflow forecasting (e.g. Yu et al., 2020; Tyralis et al., 2021a; Ni et al., 2020) and its fast implementation is beneficial in
    our spatio-temporal context. We further explore the use of the expectile loss function as a fitting criterion to give more weights
    on extremer flows. Expectile regression (Aigner et al., 1976; Newey and Powell, 1987; Kneib, 2013; Kneib et al., 2021) has
    rarely been applied in hydrology (Tyralis et al., 2022), but the use of asymmetric weights appear well suited for estimating flow
    quantiles beyond the mean (e.g. Toth, 2016). Our model shall also incorporate a variable selection procedure to obtain models
that are more parsimonious and easier to interpret.

    The objective of our study is to develop such a spatio-temporal low flow model and to evaluate its performance when predicting
    at ungauged sites. The following research questions will be addressed: (i) To what extend can spatio-temporal monthly low
    flow be modelled by one single gradient boosting model? (ii) How does expectile regression perform compared to traditional
    loss functions (mean absolute error, median absolute error)? (iii) How accurate are low extremes modeled by different expec-
tiles? (iv) Which spatial and spatio-temporal variables are used for different expectiles? Our analysis will be performed on a
    comprehensive Austrian streamflow dataset representing a range of seasonal low flow regimes.

## 2  Data and Methods

### 2.1  Data

#### 2.1.1  Hydrological Data

    Our study area covers 260 gauging stations in Austria with different low flow seasonality. The data set was already used in a
    wide range of studies (e.g. Laaha and Blöschl, 2006, 2007; Laaha et al., 2014; Laimighofer et al., 2022). Some stations included
    in previous studies had to be discarded as the gauging stations were removed, relocated or the data included too many missing
    values. The hydrological data is available from the Hydrological Service of Austria (HZB) and all stations have a continuous
daily streamflow record from 1982 to 2018. From these data we calculated the monthly low flow index series for each station,
    using the monthly Q95 ($P(Q > Q95) = 0.95$) to characterize the low flow regime (Fig. 1). The index is very similar to the
    monthly 7-day minimum flow MM(7), which has been used in an earlier study to assess the timing of low flow events on a pan-
    European scale (Laaha et al., 2017). For one station, a monthly value had to be inserted using smoothed empirical orthogonal
    functions (Lindström et al., 2014) to preserve the station despite single missing values in the daily discharge record. The
monthly Q95 was standardized by the catchment area, resulting in the monthly specific low flow (q95) time series ($ls^{-1}km^{-2}$)
    for each station, which constitutes the target variable of our study. The monthly q95 series were further transformed by the
    square root to give less weight to the high low flows, according to preliminary evaluations. Finally, model evaluation was
    performed using the predictions after transforming back to the original scale.

    Additionally we used specific discharge quantiles with 0.95, 0.98 and 0.99 exceedance probability from the daily discharge
series ($q95_d$, $q98_d$, $q99_d$) to identify extreme events in our monthly series. This threshold selection procedure classifies about

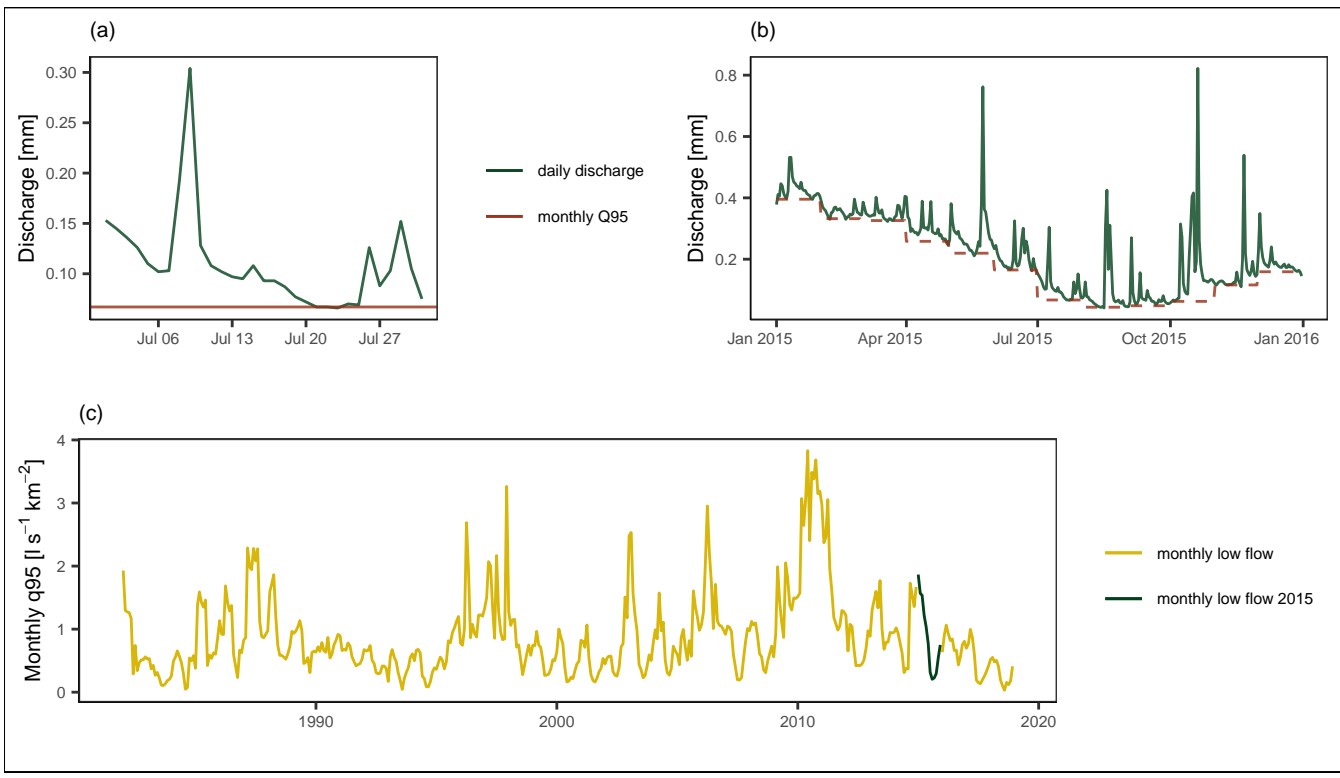

**Figure 1.** Example of the calculation of the monthly q95 time series for the gauging station Hollenstein in Lower Austria. Panel (a) shows the daily discharge in July 2015, (b) the monthly q95 and daily discharge for the full year of 2015, and (c) displays the full monthly q95 time series for the station.

11 % ($q95_d$), 5 % ($q98_d$) and 3 % ($q99_d$) of observations at each station as extreme low flow events.

### 2.1.2 Predictor variables

Spatio-temporal modeling requires predictor variables, of which two types can be distinguished. The first type consists of

climate and catchment characteristics representing the long-term average hydrological conditions. This type corresponds to typical predictors in low flow regionalisation models such as regional regression approaches (Laaha et al., 2013). The second type of predictors consists of spatio-temporal covariates that capture the climate drivers of low flow generation. These dynamic predictors are needed to extend the regional model with a temporal dimension so that space-time patterns of low flow can be represented. The former type will be referred as static predictors, the latter as spatio-temporal covariates throughout

the manuscript. For the static predictors we used a set of climate and catchment characteristics of precedent rationalisation studies in Austria (e.g. Laaha and Blöschl, 2006; Laimighofer et al., 2022). They consist of topological and landuse variables, geological classes, and static meteorological predictors. The static meteorological predictors are obtained by computing the





mean annual and seasonal aggregates of the meteorological variables such as precipitation (P, $P_w in$, $P_s um$), climate water balance (MCWB, $MCWB_w in$, $MCMB_s um$), and others (Table 1) and represent the climatic conditions of a catchment. For

more details about their calculation see Laimighofer et al. (2022) and Laaha and Blöschl (2006).

 As for the static variables, the initial choice of spatio-temporal covariates is important for data-driven models as it can affect

**Table 1.** Descriptions of static climate and catchment predictors used in the study. Abbreviations are further used in plots. Precipitation, climatic water balance, potential evapotranspiration, aridity index, snowmelt and temperature variables are used on an annual and a summer/winter half-year basis. These different accumulation periods are indicated in the subscript (no subscript for annual, win for winter, sum for summer).

| Variable | Description | Unit |
|---|---|---|
| $H_+$, $H_0$, $H_M$, $H_R$ | Maximum, minimum, mean and range of catchment altitude | m |
| A | catchment area | $km^2$ |
| Lat, Lon | Latitude and longitude of gauging station | decimal degrees |
| E | Altitude of gauging station | m |
| $S_M$ | Mean catchment slope | % |
| $S_{SL}$, $S_{MO}$, $S_{ST}$ | Fraction of slight, moderate and steep slope of the catchment | % |
| $M_S$ | Major class of fraction of slope (slight, moderate or steep) | - |
| D | Stream network density | $10^2$ m $km^{-2}$ |
| $L_U$, $L_A$, $L_C$, $L_F$, $L_G$, $L_R$, $L_W$, $L_{WA}$, $L_{GL}$ | Fraction of urban areas, agricultural areas, permanent crop, forest, grassland, wasteland, wetlands, water surfaces, glacier in catchment | % |
| $M_L$ | Major landuse class in the catchment | - |
| $G_B$, $G_G$, $G_T$, $G_F$, $G_L$, $G_C$, $G_{GS}$, $G_{GD}$, $G_{SO}$ | Fraction of bohemian massif, quaternary sediments, tertiary sediments, flysch, limestone, crystalline rock, shallow and deep groundwater table, source region in catchment | % |
| $M_G$ | Major geological class in the catchment | - |
| P | Precipitation | mm |
| $ET_P$ | Potential evapotranspiration | mm |
| AI | Aridity index | - |
| $AI_{min}$ | Half year with lower AI | - |
| MCWB | Mean climatic water balance | mm |
| S | Snowmelt | mm |
| $T_+$, $T_0$, $T_M$, $T_R$ | Maximum, minimum, mean and range of temperature | °C |
| $P_0$ | Average number of days without precipitation ($< 1$ mm) | days |
| $P_H$ | Average number of days with precipitation $> 5$ times the mean | days |

model performance and interpretability of results. In a preliminary assessment (not shown in this paper) we tested several (combinations of) spatio-temporal covariates, including various monthly aggregates of precipitation, climatic water balance,





temperature, snowmelt, solid precipitation or soil moisture. The results showed that the monthly climatic water balance (CWB)
calculated as the difference between monthly precipitation and potential evapotranspiration performs equivalently to the combination of its components provided as individual model terms. As our study focuses more on a methodological assessment we decided to include only CWB as a spatio-temporal predictor for the sake of simplicity.

The CWB enters the model as a static variable (MCWB) and on a monthly basis with different lags ($l$) from 1 to 12 month (CWB$_l$). These lags are assumed to contain information about previous months and represent antecedent conditions in low
flow generation. In addition to using "raw" CWB values, the CWB was additionally transformed in order to test whether standardization has an effect on the performance of the predictor. First, each spatio-temporal variable (CWB, CWB$_l$) is centered by month (m) and station (s) via $\mathrm{CWB}_{center,s,m} = \mathrm{CWB}_{s,m} - \overline{\mathrm{CWB}}_{s,m}$, where $\mathrm{CWB}_{s,m}$ is the monthly climatic water balance at a station, and $\overline{\mathrm{CWB}}_{s,m}$ its monthly mean. Second, we transform the climatic water balance (CWB, CWB$_l$) to a non-parametric standardized drought index (SDI), similar to the SPEI (Beguería et al., 2014). However, instead of using a parametric distribu-
tion we use the empirical probability for estimating the quantiles. A visualization of these two transformations is given in Fig. 1 and a short overview of the variables in Table 2. These two transformations are highly correlated to the initial CWB, but can provide additional information for our model. All these lags and transformations are used simultaneously as input variables for our model, resulting in 39 spatio-temporal predictor variables.

Apart from the static predictions and the spatio-temporal covariates, some variables were added to capture the temporal period-
icity. The numeric variable of the month ($m = 1, 2, \ldots, 12$), was converted to a categorical variable and additionally transformed to a sine and cosine curve so that their two-dimensional overlay gives a representation of the annual circle. In addition, the year was added as a numeric variable, and finally a cumulative sum of months ($m_{cum} = 1, 2, \ldots 444$) was added in order to represent long-term trends of low flows. In total, this results in an initial set of 116 predictor variables that are used for the variable selection procedure.


**Table 2.** Description of the different lags and transformations for the CWB. Center means centering per station and month. SDI is transforming the CWB to a standardized drought index (SDI) per station and month.

| Variable | Description | Unit |
|---|---|---|
| CWB | Monthly climatic water balance | mm |
| CWB$_l$ | Monthly climatic water balance at time lag $l$ | mm |
| CWB$_{l,center}$ | Centered CWB$_l$ with respect to station and month | mm |
| CWB$_{l,SDI}$ | Standardized drought index of the CWB$_l$ with respect to station and month | - |



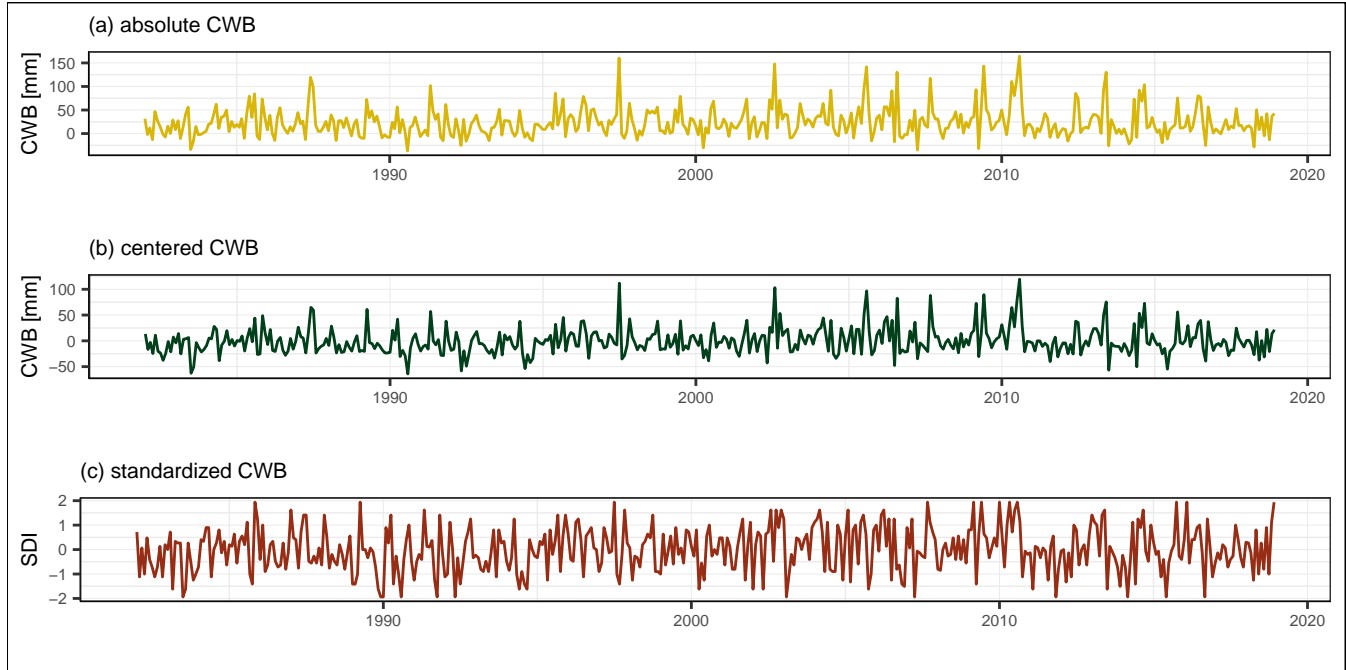

**Figure 2.** Example of the transformations of the CWB with no lag at station Hollenstein. Panel (a) shows the absolute values of the climatic water balance. (b) is the CWB centered for each month and (c) is a computation of a non-parametric SDI for each month.

## 2.2 Methods

### 2.2.1 Extreme gradient tree boosting

Extreme gradient tree boosting (Chen and He, 2015; Chen and Guestrin, 2016) is a fast implementation of gradient tree boosting, originally developed by Friedman (2001) and based on boosting algorithms in general by Friedman et al. (2000).

Gradient tree boosting consists of an ensemble of additive trees that are each fitted by a greedy algorithm to minimize a predefined loss function. Let $y$ be a vector of our response variable (monthly specific low flow q95) of length $n = s \cdot t$, where $s$ is the number of stations and $t$ the length of each individual time series per station, and index $i$ referring to its $i$-th element. Let further $X$ be our predictor matrix with $n \times p$ elements, wherein $p$ is the number of predictor variables. We can then write the regression equation as

$$\hat{y}_i = \sum_{k=1}^{K} f_k(X_i), f_k \in F, \tag{1}$$

where $f_k$ $(k = 1, \ldots, K)$ is the ensemble of regression trees, and $K$ the number of trees used. The regression trees are fitted in an additive manner, where an objective function $L^u$ is minimized:

$$L^u = \sum_{i=1}^{n} L(y_i, \hat{y}_i^{u-1} + \eta f_u(x_i)) + \Omega(f_u). \tag{2}$$





$L^u$ is the $u$-th iteration loss, $\hat{y}_i^{u-1}$ is the prediction of the regression tree in the previous iteration, $f_u$ is the tree that most

improves our model considering the predefined loss function, $\Omega(f_u)$ is an additional penalization parameter for the complexity

of the model and $\eta$ is a shrinkage parameter. To reduce the computational burden we tuned only the following hyperparameters

for the final predictions: the maximum depth (the final XGBoost was optimized for a sequence from 6 - 8) of each additive

tree, subsampling of predictor columns (0.25 - 1), which equivalently to Random Forests only uses a fraction of all predictors

to search for the optimal split and subsampling of the observations (0.5 - 1). Subsampling of the observations and the predictor

variables is used to decorrelate the trees. The maximum depth can be described as the order of interaction used in the model.

The final XGBoost model was optimized in a 10-fold CV by using all parameter combinations and tuning the number of

boosting iterations (number of trees).

### 2.2.2 Loss function

One crucial point in our study is the application of a suitable loss function. The loss function has to be a twice differentiable

convex function. Since our main aim is to model the low flows in the range of annual minima corresponding to the lower tail

of the monthly q95 series, we propose to use the expectile loss ($L_{EL}^{\tau}$) for model fitting. Expectile regression (Aigner et al.,

1976; Newey and Powell, 1987) is a squared variant of quantile regression (Koenker and Bassett, 1978), where the absolute

deviations are substituted by the squared deviations (Kneib, 2013). Expectile regression was already implemented in a boosting

framework (Sobotka and Kneib, 2012) and the resulting expectile loss can be defined by:

$$L_{EL,i}^{\tau} = \begin{cases} \tau \cdot (y_i - \hat{y}_i)^2, & (y_i - \hat{y}_i) \geq 0 \\ (1-\tau) \cdot (y_i - \hat{y}_i)^2, & (y_i - \hat{y}_i) < 0 \end{cases} \tag{3}$$

$L_{EL,i}^{\tau}$ is summed up to $L_{EL}^{\tau}$ ($L_{EL}^{\tau} = \sum_{i=1}^{n} L_{EL,i}^{\tau}$) for minimizing the error over all observations. If $\tau = 0.5$, the expectile

loss is a scaled variant of a squared loss function ($L_{squared} = \sum_{i=1}^{n} (y_i - \hat{y}_i)^2$) resulting in a least squares regression. Figure 3

shows that altering $\tau$ leads to an asymmetric weighting of the squared loss, where smaller $\tau$ give more emphasis on negative

residuals. Expectiles can not directly be interpreted as a flow quantile, as this would be possible for quantile regression, but

studies show that using transfer functions can do this very accurately (Waltrup et al., 2015) and differences mainly arise in the

tail of the distribution. Estimating a full distribution of $\tau$ values is not a very practical approach for our large dataset. Therefore,

we will assess a sequence of of $\tau$ values ($\tau = 0.01, 0.025, 0.05, 0.1, 0.2, 0.3$) and the special case of a least squares regression

($\tau = 0.5$). Within this sequence, the $\tau$ values of $0.1, 0.05$ and $0.025$ give good approximations of our three thresholds: q95$_d$,

q98$_d$ and q99$_d$. Finally, we will compare the expectile loss function to the absolute loss (Fig. 3):

$$L_{abs,i} = |y_i - \hat{y}_i|. \tag{4}$$

In case of the absolute loss, the mean absolute error will be minimized by the mean ($L_{MAE} = \frac{\sum_{i=1}^{n} L_{abs,i}}{n}$), and in case of the

median absolute loss by the median ($L_{MDAE} = median(L_{abs,i})$). In both cases the second derivative is approximated by a

vector of 1s.



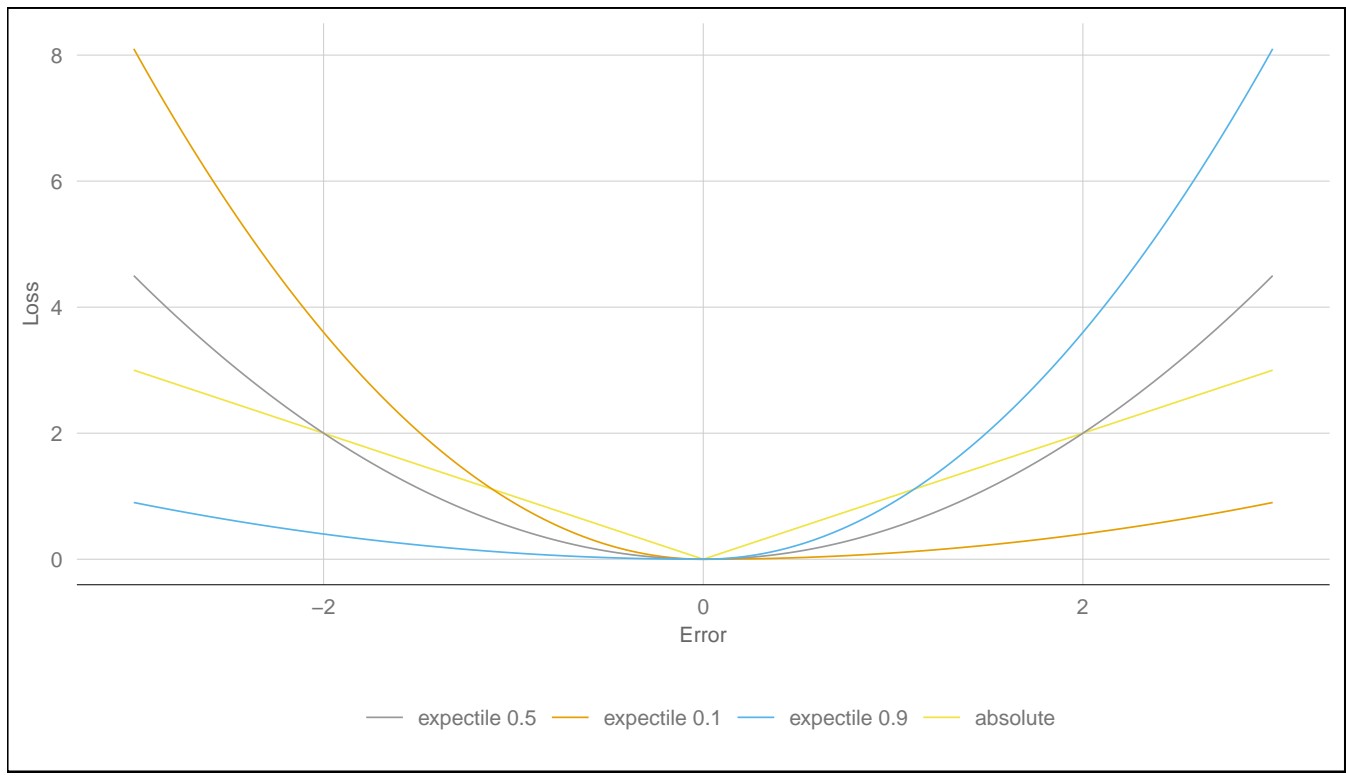

**Figure 3.** Comparison of different loss functions. Shown are expectile loss functions for various $\tau$ parameters and the absolute loss function.

### 2.2.3 Variable selection

Model evaluation is performed by a nested 10-fold CV (Varmuza and Filzmoser, 2016). A nested CV contains two loops, an inner loop which is used for tuning of hyperparameters and variable selection and an outer loop for evaluating the predictive performance of the models (Laimighofer et al., 2022). In each inner loop, we include a variable selection by recursive feature elimination (RFE, Granitto et al., 2006). The RFE algorithm consists of an initial variable ranking and a backward variable selection. The initial variable ranking is computed by using the XGBoost algorithm with 500 boosting iterations and default hyperparameters (maximum depth = 8, subsample = 1, fraction of p = 0.5). The variables are ranked after their additive gain in minimizing the loss function over the 500 boosting steps. For a more robust approach, the initial variable ranking is averaged over 25 bootstrap samples. In a next step, we are fitting a XGBoost model to a sequence from 5 to the maximum number of variables ($p$), but only with a step length of 5 to reduce the computational burden. For each number of variable the error is calculated and averaged over all 10 CV runs of the inner loop. The number of variables are then determined by using a threshold of 1.05 times the minimum error to produce parsimonious models. The applied method is fully described in Laimighofer et al. (2022) and to increase clarity we added a visual explanation of the full cross validation procedure (Fig. 4).





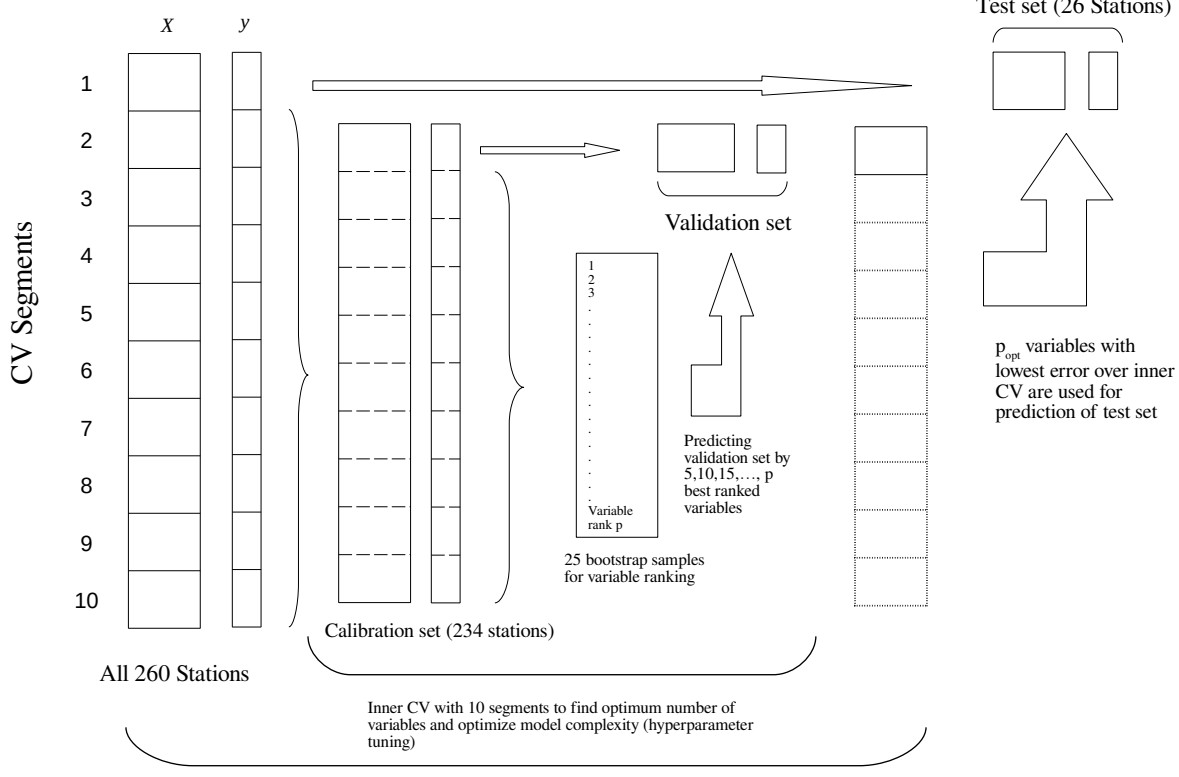

**Figure 4.** The nested CV procedure as adapted from the double CV-scheme of Varmuza and Filzmoser (2016).

### 2.2.4  Model evaluation

Model evaluation is performed by several error metrics. First, we quantify the overall performance of the model by means of
four error metrics, the median absolute error ($MDAE = L_{MDAE}$), the mean absolute error ($MAE = L_{MAE}$), the root mean
squared error (RMSE):

$$RMSE = \sqrt{1/N \sum_{i=1}^{N} (y_i - \hat{y}_i)^2},$$ (5)





and the coefficient of determination $R^2$:

$$R^2 = 1 - \frac{\sum_{i=1}^{N}(y_i - \hat{y})^2}{\sum_{i=1}^{N}(y_i - \overline{y})^2}. \tag{6}$$

All measures are based on cross-validation and therefore indicators of the predictive performance when estimating at ungauged sites.

Second, we assess the predictive performance for individual stations, by calculating the $R^2$ for each station separately. The station-by-station performance is summarized by the distribution and the median $R^2$ over all stations, which will be referred as $R^2_{med}$ throughout the manuscript.

For a more comprehensive evaluation of the performance, we perform a decomposition of the station-wise prediction errors on different time scales. The purpose is to assess to what extend the model errors occur at the annual, seasonal, and monthly level, and which part of the error is due to a systematic error (i.e. bias). This will allow us to get insight in structural strength and weaknesses of the models. This decomposition is done by a three-way ANOVA, which was applied, for example, by Parajka et al. (2016) for assessing uncertainty contributions of climate projections. The basic linear model for our ANOVA can be defined as:

$$y_{m,yr} - \hat{y}_{m,yr} = \mu + \alpha_m + \beta_{yr} + \epsilon_{m,yr}, \tag{7}$$

where the left-hand side (predictand) are the prediction errors of a model at a single station. These are decomposed into the terms $\mu$ representing the mean error (i.e. bias), $\alpha_m$ the seasonal effects ($m = 1, 2, \ldots, 12$), $\beta_{yr}$ the mean annual effects ($yr = 1, 2, \ldots, 37$), and the residual term $\epsilon_{m,yr}$ corresponding to the model errors at the monthly level. Note that $\mu$ is not a classical intercept but enters as constant factor (coded as a vector of 1s) which allows the bias to be considered as a separate effect in the error decomposition. In the ANOVA framework, the total variance of the prediction errors is characterised by the total sum of squares $SS_T$, and is decomposed into additive variance components of individual effects:

$$SS_T = SS_{bias} + SS_{season} + SS_{year} + SS_E. \tag{8}$$

The variance contributions of each term are estimated by the measure $\omega^2$, which is an analogue to the coefficient of determination. The measure $\omega^2$ of, e.g., the seasonal effect is defined as:

$$\omega^2_{season} = \frac{SS_{season} - df_{season} \cdot MS_E}{SS_T - MS_E}. \tag{9}$$

where $df_{season}$ are the degrees of freedom (i.e. number of factor levels - 1), and $MS_E = SS_E/df_E$ the residual mean squared error, which can be seen from the ANOVA table. The contribution of the mean annual effect ($\omega^2_{year}$) and the fraction of the bias ($\omega^2_{bias}$) can be determined analogously and a more detailed description can be found in (Parajka et al., 2016). Finally, the residual term $\omega^2_E$ can be calculated as:

$$\omega^2_E = 1 - \omega^2_{season} - \omega^2_{year} - \omega^2_{bias}. \tag{10}$$


The last part of our model evaluation gives a specific emphasis on the extreme low flows. This is assessed by three performance metrics. First, we filter the observations at each station by a specific quantile and calculate the overall $R^2$ based on the residuals of the filtered cases. Second, we calculate the relative expectile error ($EL_\tau$) at each station:

$$EL_\tau = 1 - \frac{L_{EL}^\tau(y_i, \hat{y}_i)}{L_{EL}^\tau(y_i, q(y, \tau))}. \tag{11}$$

The error metric is defined in analogy to the expectile loss function to give more weight to the fraction of $\tau$ lowest values. The denominator is the ($L_{EL}^\tau$) of a naive estimate for each station. For this purpose, we are using the $\tau$-quantile of the observations ($q(y, \tau)$) as simple prediction at the specific station, assuming that quantiles and expectiles give similar estimates. The numerator, on the other hand, is the expectile error ($L_{EL}^\tau$) for the predictions of our model. Similar to the $R^2$, the $EL_\tau$ is bounded at a

maximum of 1, representing perfect model fit. Values below 0 indicate that the naive prediction of the $\tau$-quantile would result in a better prediction than our model. One advantage of this metric is that it is based on the full data, but gives more weight to the values below our selected $\tau$.

Finally, we want to capture the model ability of classifying extreme low flow events. For this purpose, the specific low flow quantiles q95$_d$, q98$_d$ and q99$_d$ are calculated from the daily discharge records and used as thresholds for identifying drought

events in the monthly low flow series. Based on these thresholds, the drought / no drought cases of observed and predicted monthly time series are binary coded, and we calculate the hit score ($H_S$) and the precision ($P_{rec}$) for evaluating the performance at each station. The hit score (also termed recall, sensitivity or true positive rate) is calculated by:

$$H_S = ET_{pred}/E_{obs}, \tag{12}$$

where $E_{obs}$ are the number of low flow events in the originally data and $ET_{pred}$ is the number of low flow events correctly

classified by the predictions. The precision is computed by:

$$P_{rec} = \frac{ET_{pred}}{ET_{pred} + EF_{pred}}, \tag{13}$$

where $EF_{pred}$ is the number of low flows falsely predicted by each monthly time series.

The data analysis was performed in R (R Core Team, 2021) and we want to acknowledge the use of the following packages: XGBoost (Chen et al., 2021), dplyr (Wickham et al., 2021), purrr (Henry and Wickham, 2020), tidyr (Wickham, 2021), ggplot2

(Wickham, 2016), caret (Kuhn, 2021), glmnet (Friedman et al., 2010), tidyselect (Henry and Wickham, 2021), tibble (Müller and Wickham, 2021), receipes (Kuhn and Wickham, 2021), ggthemes (Arnold, 2021), lubridate (Grolemund and Wickham, 2011), wesanderson (Ram and Wickham, 2018).

## 3 Results

### 3.1 Global model performance

Table 3 presents the results for the overall performance of our spatio-temporal model. Generally, most loss functions show a good overall performance. The 0.5 expectile yields the best $R^2$ of 0.81, which is substantially higher than 0.73 for the





median absolute loss and 0.74 for the mean absolute loss. Regarding the MDAE, which gives more focus on low values, there is no change in the overall ranking of methods, with a MDAE of 1.81 for the 0.5 expectile, 1.99 for the median absolute loss and 2.01 for the mean absolute loss. Regarding the results of the expectiles with $\tau$ smaller than 0.5, we can identify a

decreasing performance towards smaller expectiles over all performance metrics. The $R^2$ is stable until the 0.3 expectile and than suddenly drops from 0.8 (0.3 expectile) to 0.2 for the 0.01 expectile. A similar loss in performance can also be observed with the MDAE, MAE or RMSE. Nevertheless, expectiles as 0.2 and 0.3 show better overall metrics than the median absolute loss and the absolute loss and even the 0.1 expectile demonstrates only a somewhat lower $R^2$ of 0.7.

**Table 3.** Overview of the error metrics for all loss functions. Shown are the MDAE, MAE, RMSE and $R^2$, all representing the overall predictive performance of the model for the study area.

| Loss function | MDAE | MAE | RMSE | $R^2$ |
|---|---|---|---|---|
| Absolute | 2.01 | 3.95 | 8.28 | 0.74 |
| Median absolute | 1.99 | 3.95 | 8.42 | 0.73 |
| Expectile 0.5 | 1.81 | 3.50 | 6.98 | 0.81 |
| Expectile 0.3 | 1.85 | 3.58 | 7.20 | 0.80 |
| Expectile 0.2 | 1.90 | 3.80 | 7.72 | 0.77 |
| Expectile 0.1 | 2.09 | 4.26 | 8.77 | 0.70 |
| Expectile 0.05 | 2.72 | 5.61 | 11.21 | 0.52 |
| Expectile 0.025 | 3.36 | 6.63 | 12.87 | 0.36 |
| Expectile 0.01 | 4.21 | 7.79 | 14.45 | 0.20 |

### 3.2 Station-by-station performance

A more detailed examination of our results is realized by analyzing the performance per station. Table 4 gives an overview of the results and Fig. 5 shows the empirical cumulative distribution of the station-specific $R^2$. From the graphs, the 0.5, 0.3 and 0.2 expectiles outperform the other models across all stations. The 0.5 expectile has the highest $R^2_{med}$ (0.67) and shows a far better performance than the median absolute loss (0.57) and the mean absolute loss (0.58). This generally good performance is further underlined by the finding that only 26 % of the stations have a $R^2$ below 0.5 for the 0.5 expectile. In contrast, for

the mean and median absolut loss about 40 % of the stations yield a $R^2$ below 0.5. For the expectiles lower than 0.5 we can identify a decrease of the $R^2_{med}$, from 0.65 for the 0.3 expectile to 0.14 for the 0.05 expectile. The 0.01 and 0.025 expectile yield a negative $R^2_{med}$ and thus a high number of inadequate models. However, the 0.2 and 0.3 expectile still show better performance than the mean absolute and median absolute loss in terms of the $R^2_{med}$ and also a low portion of stations with weak performance (only 36 % (0.2) and 32 % (0.3) stations have a $R^2$ below 0.5). These findings suggest, in response to our first

research question, that a single model can provide very accurate results for most stations ($0.5, 0.3, 0.2$ expectile), but 26 % to 36 % of the stations have inadequate performance. The origin of these errors will be analysed in more depth in the subsequent section. As the results of the mean and the median absolute loss could not compete with the performance of the $0.2, 0.3, 0.5$





**Figure 5.** Empirical cumulative distribution function of station-wise $R^2$ by loss function across all stations. For improving visual clarity the $x$-axis is bounded at -1.

expectiles, we will not further include these two loss functions in our analysis.

### 3.2.1 Error decomposition

For a better understanding of the error at the individual stations we decompose the model error at each station into monthly, seasonal, annual fractions and a component representing the average error (prediction bias). Figure 6a gives an overview of the error components. For all expectiles, $\omega_{bias}^2$ and $\omega_E^2$ are the most important components. For the expectiles from 0.1 to 0.5 the $\omega_E^2$ is the main error contribution with median values between 52 % (0.1 expectile) and 59 % (0.2 expectile). The median values of $\omega_{bias}^2$ increases with decreasing $\tau$, from 15 % to 19 % for the 0.5 to 0.2 expectiles, and 28 % for the 0.1 expectile. This trend continues for the smaller expectiles where the fraction of the mean error rises up to 56 % for the 0.01 expectile. At





**Table 4.** Performance per station summarised by the median $R^2$ over all stations ($R^2_{med}$) and the fractions of stations with an $R^2$ below 0.5.

| Loss function | $R^2_{med}$ | $R^2 < 0.5$ |
|---|---|---|
| Absolute | 0.58 | 0.39 |
| Median absolute | 0.57 | 0.40 |
| Expectile 0.5 | 0.67 | 0.26 |
| Expectile 0.3 | 0.65 | 0.32 |
| Expectile 0.2 | 0.60 | 0.36 |
| Expectile 0.1 | 0.48 | 0.54 |
| Expectile 0.05 | 0.14 | 0.89 |
| Expectile 0.025 | -0.13 | 0.98 |
| Expectile 0.01 | -0.50 | 1.00 |

the same time $\omega^2_E$ decreases for smaller expectiles, showing that expectiles with small $\tau$ are better fitted to the monthly values but predictions are generally less accurate. With median values around 10 % and between 3 and 6 %, respectively, the seasonal and annual errors are much smaller than the mean error component and show a good performance of the models in predicting annual and seasonal low flow variability.

An additional perspective comes from analyzing the share of stations showing only moderate or weak performance, as indicated by an $R^2$ of less than 0.5. As an example, Fig. 6b shows such ill-performing stations for the case of the 0.5 expectile model, with 67 stations (or 26 %) having an $R^2$ below 0.5. The main error contribution for these stations is the $\omega^2_{bias}$ with a median value of 56 %, which is much higher than the $\omega^2_{bias}$ of the well-performing stations (median value of 11 %). This again underlines the earlier finding that the main shortcoming of our modeling approach is an error in the mean, which reduces the predictive accuracy of the models. Seasonal and monthly variability, though, is well covered by the models, which is a strength of our spatio-temporal modeling approach.

### 3.3 Prediction of extremes

One objective of this study was the application of different expectiles for improving predictions at low extremes. This section will consequently focus on their evaluation. In a first step we will filter our observations by considering only the observations at each station below a specific quantile. The filtered observations are then used to calculate an overall $R^2$, which is shown in Fig. 7. We can identify a huge performance advantage for low expectiles for predicting low extremes. For example if we assess the accuracy of our models for cases below the 1 % quantile, the 0.01 expectile is yielding a $R^2$ of 0.42, where larger expectiles (0.1 - 0.5) show inefficient models with a $R^2$ below 0. What becomes apparent is that the performance is dropping at some point for the low expectiles (0.01, 0.025, 0.05), but is monotonically increasing for all other expectiles. Considering the results at the 25 % quantile, the $R^2$ of the 0.01 expectile decreased to 0.28, so that the best performance is at the 0.1 (0.53) and 0.05 expectiles (0.51). Further we found that the 0.5 expectile has a lower performance than the 0.1, 0.2 and 0.3 expectiles

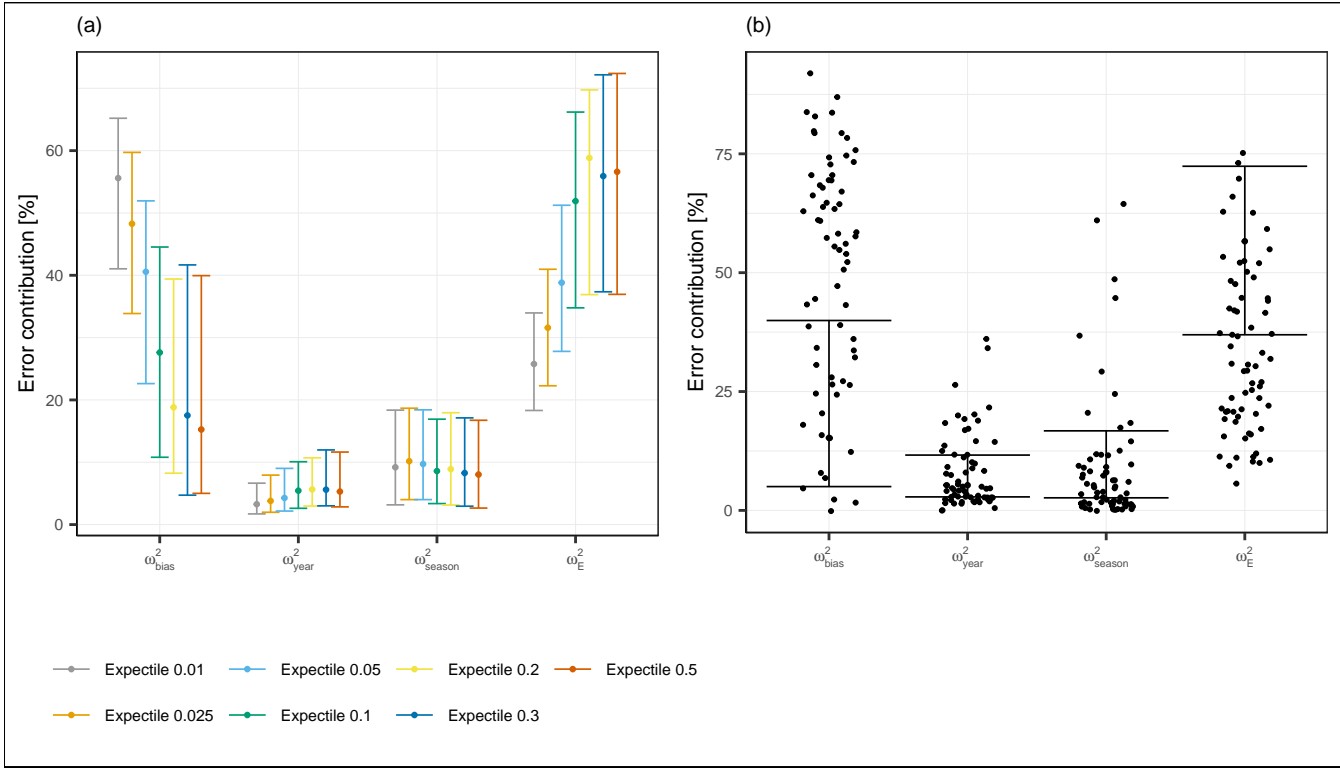

**Figure 6.** Error components of various expectile models. Panel (a) shows the relative error contributions for the season, year, month and the bias part across all stations. Shown are the median (point) together with the 25 % quantile and the 75 % quantile (whiskers) of relative errors for each expectile loss function. Panel (b) displays the error components for the 0.5 expectile in greater detail. The points are the stations which have a $R^2$ lower than 0.5.

even for estimating the lower 50 % of the data.

As a second assessment of the predictive performance for low extremes, we compute the relative expectile error ($EL_\tau$), a weighted error metric that distributes weights by expectile functions that have greatest weight around $\tau$. We choose $\tau$ values of $0.1, 0.05$ and $0.025$, as these approximately correspond to the q95$_d$, q98$_d$ and q99$_d$ thresholds used for drought identification. Figure 8 shows the expectile errors $EL_\tau$ calculated for every station. Generally, the $0.1$ expectile demonstrates a good performance by all $EL_\tau$ metrics. Smaller expectiles $(0.01, 0.025, 0.05)$ perform as good or better than the $0.1$ expectile for the
$EL_{\tau=0.025}$ (which gives most weight to most extreme events), but not for the other $EL_\tau$ metrics. Interestingly, the expectiles used for optimizing our loss functions do not perform best considering the respective $EL_\tau$ metric. For example, we would assume that the best performing expectile for the $EL_{\tau=0.025}$ would be the $0.025$ expectile, but on median this expectile yields a $EL_{\tau=0.025}$ of $0.39$, which is somewhat lower than the $0.1$ expectile $(0.4)$ or the $0.025$ expectile $(0.44)$. Further, the smallest expectile $(0.01)$ only obtains a median $EL_{\tau=0.025}$ of $0.29$, which is only slightly higher than the $0.2$ expectile $(0.28)$. Higher
expectiles $(0.2, 0.3, 0.5)$ show an increasing performance with higher $\tau$, but their performance varies more across all stations.



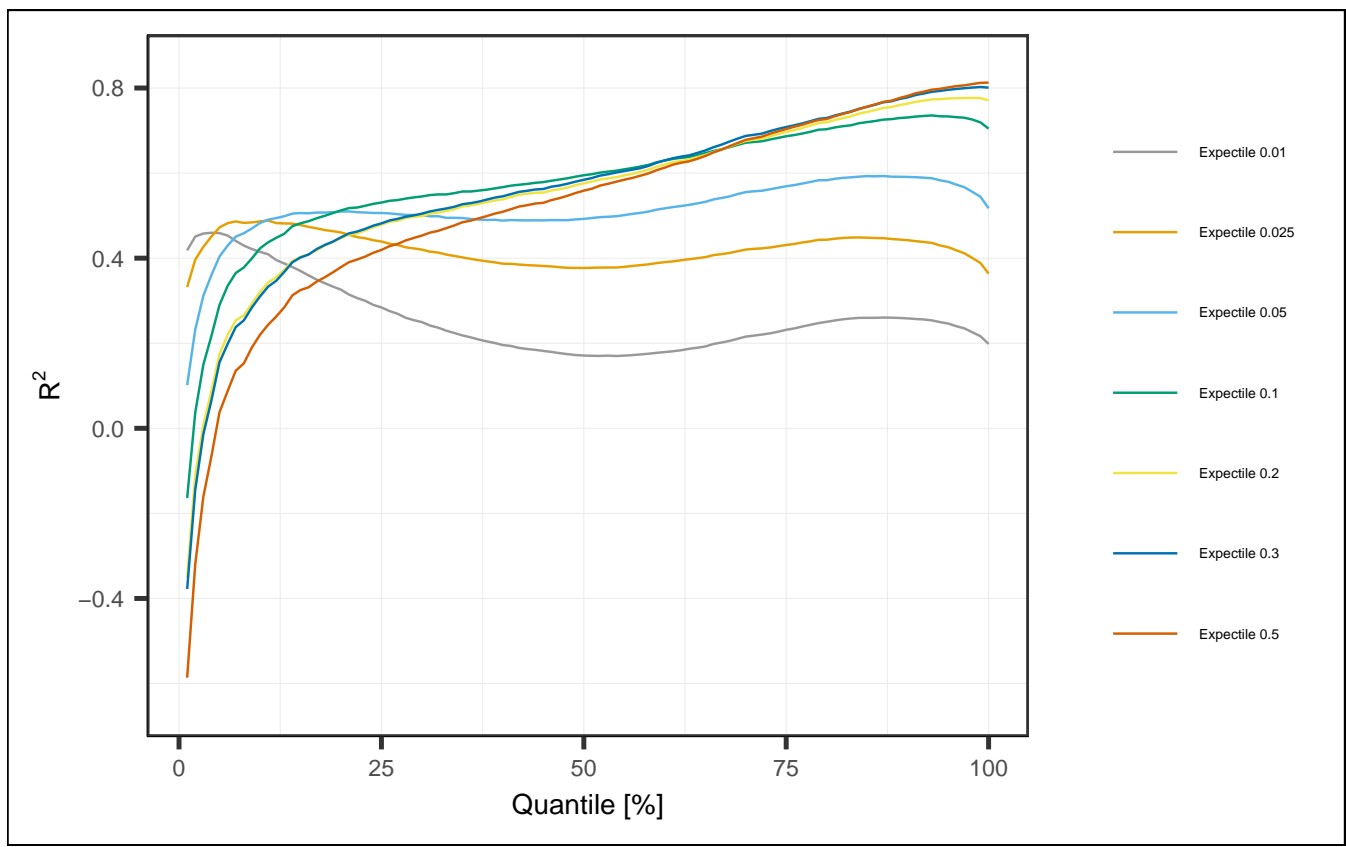

**Figure 7.** Global $R^2$ conditional to observations below a low flow thresholds. The thresholds are (station-wise) quantiles of the monthly time series with 1 % to 100 % non-exceedance probability.

This suggest that one should choose an expectile function that gives most weight to somewhat higher values than the drought threshold of interest, and $\tau$ values around 0.1 appear most accurate for q95$_d$, q98$_d$ drought events.

In a final assessment of the model performance, we analyse the skill of the models to classify extreme events. For this purpose we focus on the hit score ($H_S$) and the precision ($P_{rec}$) at each station. Table 5 shows the median of these two metrics over
all stations and for all expectiles. Both skill scores decrease from less extreme (q95$_d$) to more extreme (q99$_d$) low flows for all expectiles. Concerning the effect of expectiles on $P_{rec}$ and $H_S$, we see an opposite behaviour, with $H_S$ increasing towards the lower expectiles, and $P_{rec}$ decreasing. For the lowest expectile (0.01), the $H_S$ reaches 0.98, 0.94 and 0.9 for the three low flow thresholds, respectively, showing that almost every observed drought event is identified. On the other hand the $P_{rec}$ is 0.2 for the q95$_d$ and only 0.07 on median for the q99$_d$, which means that only 7 % of the predicted extreme events per station were
actually extreme events. This points to an underestimation of extreme low flows when using low expectiles for model fitting. In contrast, using the 0.5 expectile would lead to an increase in the precision rate, but to a very low hit score. Fig. 9 shows the contrasting properties of a high and a low expectile model on low flow predictions. The low expectile shifts the predictions to





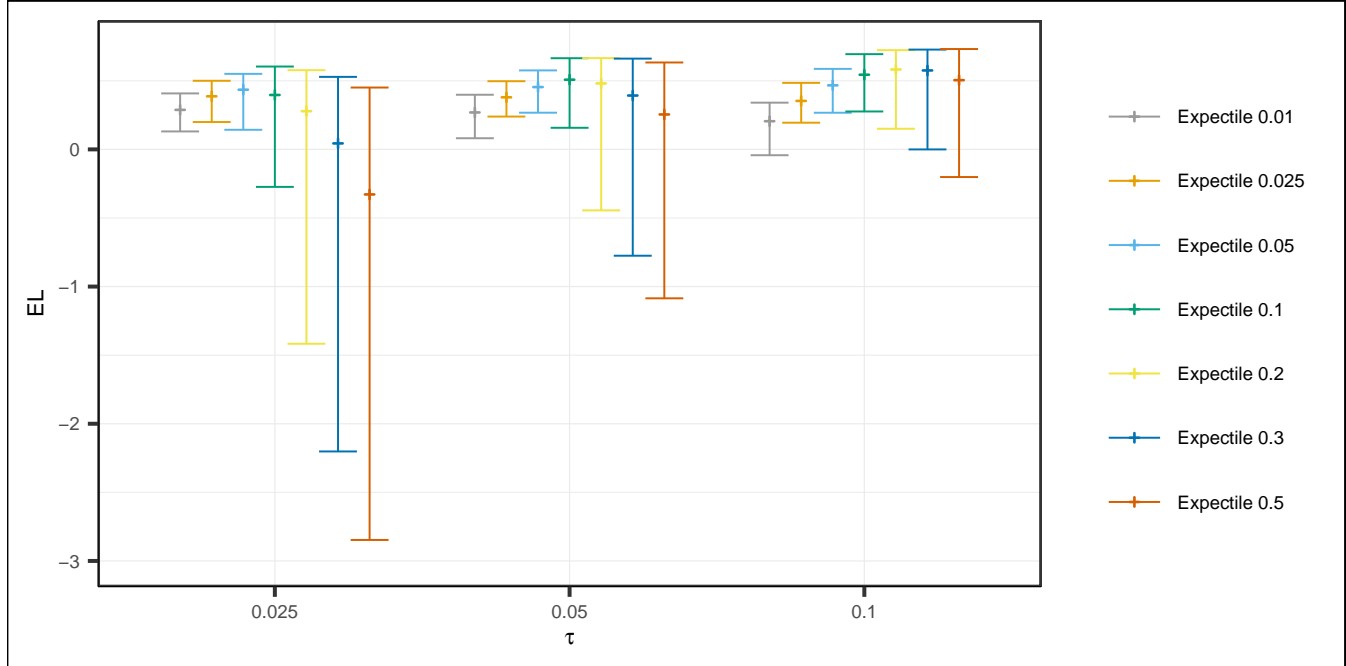

**Figure 8.** $EL_\tau$ for $\tau = 0.025, 0.05, 0.1$ presented for all expectiles. The median, 25 % quantile and 75 % quantile over all stations and each expectile are displayed.

match the most extreme events, but underestimates the moderate events. Clearly, such low expectiles result in strongly biased models with little practical relevance. These findings suggest a trade-off between accurate prediction of extremes, overall pre-

diction accuracy and correct classification of extreme events, where the user needs to find some optimum.

**Table 5.** The precision $P_{rec}$ and hit rate $H_S$ are computed per station and the median is shown in the table for $q95_d$, $q98_d$ and $q99_d$.

|  | q95$_d$ | | q98$_d$ | | q99$_d$ | |
| :--- | :---: | :---: | :---: | :---: | :---: | :---: |
| Loss function | $P_{rec}$ | $H_S$ | $P_{rec}$ | $H_S$ | $P_{rec}$ | $H_S$ |
| Expectile 0.5 | 0.61 | 0.50 | 0.37 | 0.33 | 0.21 | 0.18 |
| Expectile 0.3 | 0.59 | 0.54 | 0.41 | 0.37 | 0.21 | 0.24 |
| Expectile 0.2 | 0.53 | 0.59 | 0.35 | 0.44 | 0.21 | 0.29 |
| Expectile 0.1 | 0.49 | 0.72 | 0.30 | 0.52 | 0.16 | 0.36 |
| Expectile 0.05 | 0.32 | 0.84 | 0.19 | 0.65 | 0.10 | 0.45 |
| Expectile 0.025 | 0.26 | 0.93 | 0.15 | 0.82 | 0.10 | 0.64 |
| Expectile 0.01 | 0.20 | 0.98 | 0.11 | 0.94 | 0.07 | 0.90 |

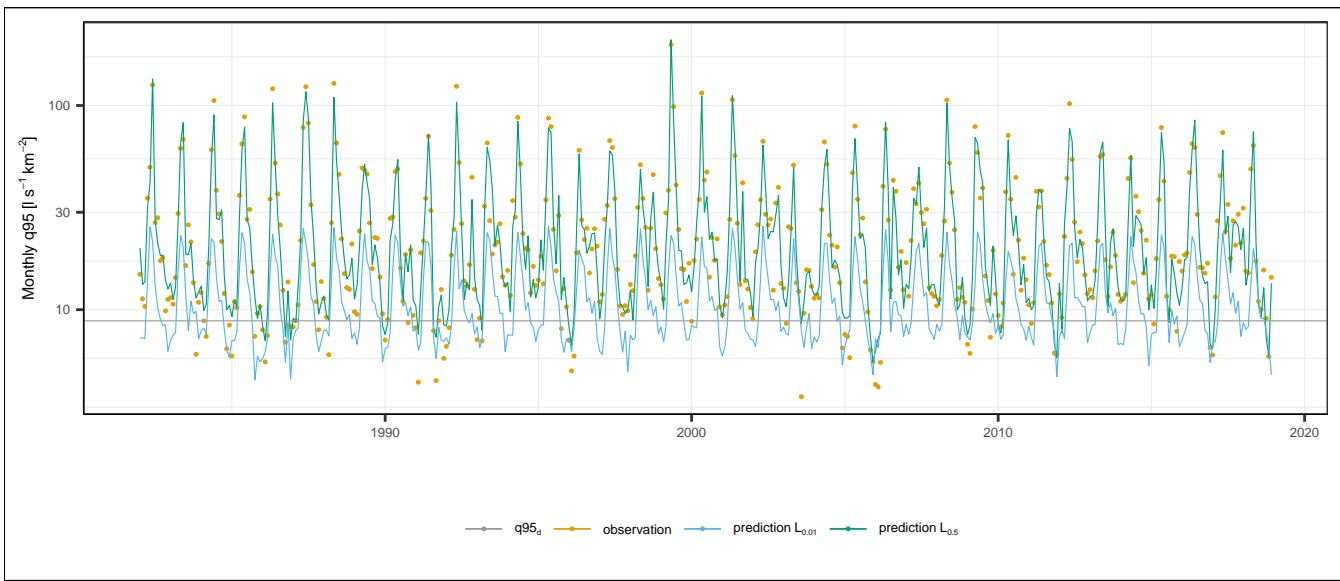

**Figure 9.** The predictions for the $0.01$ and $0.5$ expectile are shown for station Au/Bregenzerach. The $y$-axis is on a log-scale for better visualization of the low extremes. Error metrics for the $0.5$ expectile: $R^2 = 0.95$, $H_S = 0.51$, $P_{rec} = 0.79$; $0.01$ expectile: $R^2 = -0.07$, $H_S = 0.94$, $P_{rec} = 0.25$.

### 3.4 Variable selection

The implemented RFE algorithm leads to a substantially reduction of variables. On median, the different expectiles require between 20 and 30 variables, except the $0.5$ expectile has a median of 35 variables. A closer examination of the static predictors

(an overview of the selection of the static variables is given in Appendix A) shows that geological variables are never selected for any expectiles, and also from the landuse variables only the fraction of forest is selected frequently. However, a larger number of meteorological variables were selected over all expectiles: aridity ($AI$, $AI_{summer}$), annual climatic water balance ($CWB_{annual}$), annual precipitation ($P_{annual}$) and days with zero precipitation in the summer months ($P_{0,summer}$). Despite their frequent appearance in models, the performance gain of using meteorological variables for predictions is low, with a

typical variable importance of less than 2 % for all models. In contrast, topological variables emerge as the dominant predictors in the model. The three topological variables yielding the largest performance gain in the models are the mean and maximum catchment altitude ($H_M, H_+$) and the mean catchment slope $S_M$. The variable importance for $H_M$ is on average between $3.8$ % and $4.4$ % for all expectiles. A slightly increasing trend towards larger expectiles can be observed for $S_M$, with a variable importance of $10$ % for the $0.01$ expectile and on average $11.6$ % for the $0.5$ expectile. An opposed effect can be

observed for $H_+$, where the variable importance decreases from $7.2$ % for the $0.01$ expectile to $4.3$ % for the $0.5$ expectile. Other topological predictors whose inclusion had only a minor influence on model performance are catchment area, latitude, longitude and altitude of the gauging station, stream network density, and the fractions of flat, moderate and steep slopes in the catchment.



**Figure 10.** Selection of spatio-temporal variables that are used for different expectiles. For each expectile it is shown if the variable is used for the final prediction of the fold or not.

Figure 10 shows an overview of the variable importance for the most relevant spatio-temporal predictors. The CWB is the
most important variable with more than 2 % performance gain over all expectiles. $CWB_{center}$, $CWB_1$, $CWB_{1,center}$ are
also included in all models, but their variable importance is only 1 to 1.5 %. Higher lags of the $CWB$ are mainly included by
the lower expectiles but their performance gain is somewhat smaller and decreases with the lag of the variable. Comparing raw
and standardized climate variables, the transformation of the absolute values to centered values seem to be beneficial in the
case of the climate water balance, but this is not the case for the standardized drought indices, which were rarely selected and
exhibited a low importance score in all cases.



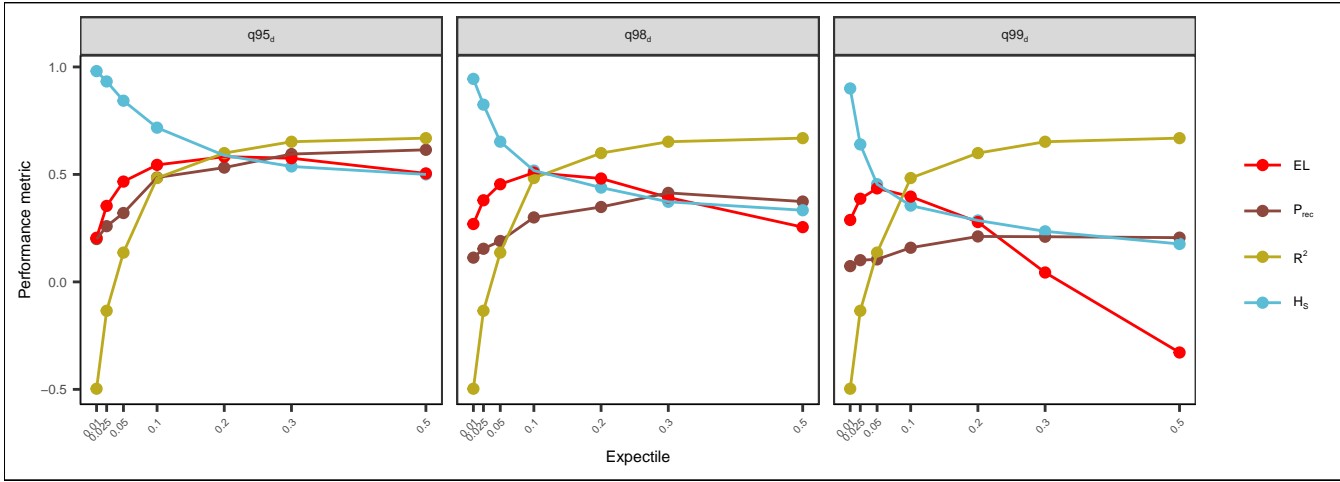

**Figure 11.** Overview of the error metrics $EL_\tau$, $P_{rec}$, $H_S$ and $R^2_{med}$, plotted against every expectile. $H_S$ and $P_{rec}$ are computed in each panel for the thresholds $q95_d$, $q98_d$ and $q99_d$. The $EL_\tau$ is calculated for the $\tau$ values of 0.025 ($q95_d$), 0.05 ($q98_d$) and 0.1 ($q99_d$).

## 4 Discussion

### 4.1 Value of expectile regression tree models for low flow estimation

In this paper we extended the extreme gradient boosting model XGBoost with the expectile loss function to develop a space-
time model for low flow predictions. We applied different $\tau$ values from 0.01 to 0.5 and evaluated their predictive quality
in terms of overall performance, accuracy at extremes and potential to classify extreme events in a time series. Our findings
showed two contrasting behaviours. Larger expectiles as 0.2, 0.3 and 0.5 showed better performance in respect to $R^2$ and $R^2_{med}$,
whereas low expectiles ($\tau = 0.01, 0.025, 0.05$) led to a sharp decline in $R^2_{med}$ but increasing prediction accuracy at extreme low
flows. However, optimization of the model can not be reduced to these two criteria, as was shown in the further assessments.
Low expectiles led to a reduced precision at extreme low flow events, as a result of too many events being predicted. This
suggests an overfitting of the lower tail of the low flow distribution by low expectiles that reduces the predictive performance
at the entire distribution. Therefore, the application of the expectile loss has to be considered carefully and adjusted to the
specific research question. Figure 11 gives a synopsis of key performance metrics with respect to predicting $q95_d$, $q98_d$ and
$q99_d$ low flow events. The synoptic representation shows nicely the trade-off between different performance metrics and puts
the conclusions from their individual assessment into context. Depending on the low flow event a different optimum can be
observed. When the focus of the study is on annual low flows in the order of $q95_d$, we see that the 0.2 expectile yields the
optimal model fit, indicated by the crossing performance lines of the various metrics. However, when the purpose of the study
is on predicting more extreme events ($q98_d$ and $q99_d$) the 0.1 expectile is the optimal choice. These optima take into account
the overall predictive performance ($R^2_{med}$) and emphases the performance at the considered extreme, by the expectile loss, the
hit rate and precision at the event of interest. The optimized model shows a good performance in all metrics and appears well



suited for spatio-temporal predictions of low flow events.

## 4.2 Performance compared to literature

Performance evaluation in light of the existing literature is not straightforward, as we are not aware of any studies evaluating
monthly low flow models. Nevertheless, some studies did model the mean monthly streamflow, which we will use for com-
parison. Comparisons will be made on the Nash–Sutcliffe efficiency (NSE) reported in these studies, which is an analogue
measure to the $R^2$ in our study (Blöschl et al., 2013; Parajka et al., 2013). To put our monthly q95 models into context, we
run a XGBoost model on monthly mean flow using the mean expectile regression (0.5) for our dataset. This leads to a median
performance $R^2_{med}$ of 0.77 with only 15 % of the stations having an $R^2$ below 0.5. As expected, modelling monthly low flow is
less performant than modeling the mean monthly streamflow. Nevertheless, our results for q95 are still in the range of published
studies on mean monthly streamflow models. For example Cutore et al. (2007) found NSE values ranging from 0.57 to 0.78
by modelling 9 basins in Italy using artificial neural networks. A similar performance has been reported by Pumo et al. (2016)
for 59 stations in Sicilia, Italy. They modeled monthly streamflow by interpolating regression parameters to ungauged basins
and their NSE values ranged from 0.7 to 0.8 for 6 selected validation stations. A comparative assessment of regionalization ap-
proaches for hydrological models for 22 stations in the USA (Steinschneider et al., 2015) showed NSE values ranging from 0.6
to 0.85. Note that these results were obtained for relatively small datasets under quite homogenuous hydrological conditions.
In our study low flow predictions were evaluated on a larger, diverse hydrological dataset and we found similar performance
metrics to those in the aforementioned studies.

A more qualitative embedding to the scientific literature can be made by integrating our findings to the comparative evaluation
of regionalization procedures of Parajka et al. (2013). Parajka et al. (2013) evaluated the performance of runoff-hydrograph
studies, mostly performed on a daily time step, as a function of aridity, elevation and catchment area to assess to what ex-
tent does performance depend on main climate and catchment characteristics. Figure 12 shows our finding in context of these
three catchment characteristics. Parajka et al. (2013) found a decreasing performance with higher aridity. In our study, the
aridity index is below 0.6 for all our catchments, but shows a similar decrease in performance. In our study the performance
increases with catchment elevation, with catchments below 450 m.a.s.l. having the lowest performance ($R^2_{med}$ of 0.5) and
catchments above 1350 m.a.s.l the highest performance ($R^2_{med}$ of 0.8). The increase in performance with elevation points to a
higher performance for catchments with dominant winter low flow seasonality, which can be explained by their more regular
and thus better predictable regime. We can indeed observe a lower performance for stations with summer low flow regime
($R^2_{med} = 0.63$) than for stations with winter low flow regime ($R^2_{med} = 0.73$). However, in a stratified assessment of summer
and winter months, we could not find any difference in performance, neither in the entire study area, nor for summer and winter
dominated catchments separately. This is in contrast to Staudinger et al. (2011), who found a reduction in model performance
for summer months compared to winter months for a Norwegian study area with conditions comparable to the mountainous
catchments in our study. Finally, we also found more accurate model predictions for larger basins. These are not due to a simple
size effect of catchment area on discharges as we use specific low flows as target variable to make catchments of different size



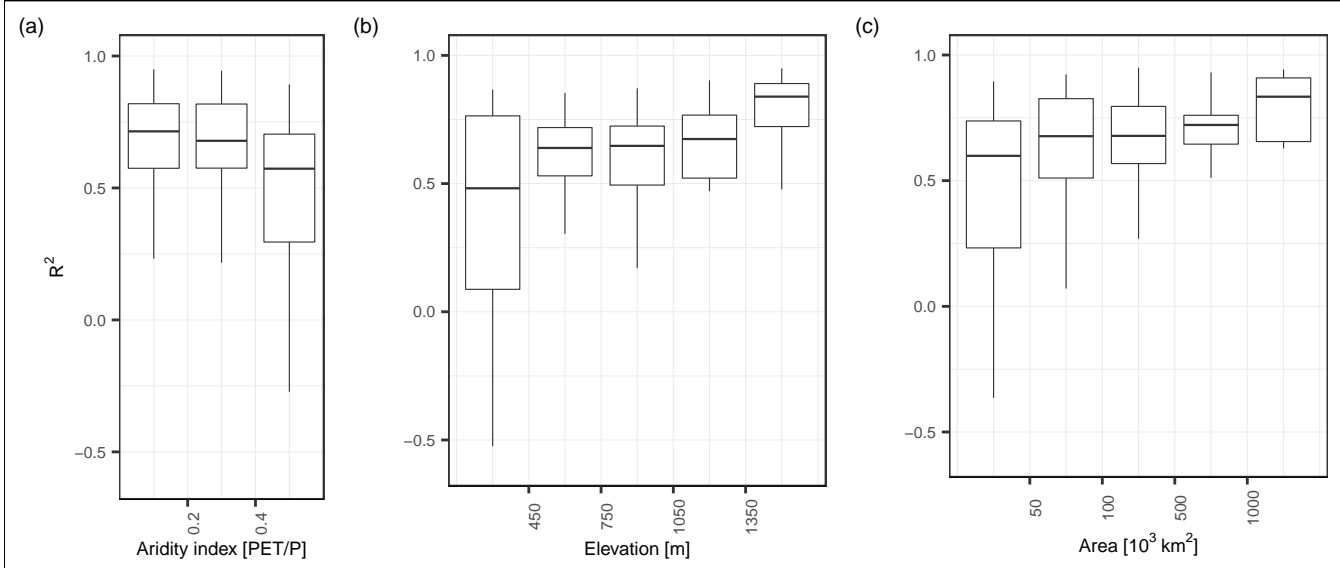

**Figure 12.** Panels (a), (b) and (c) show $R^2$ per station of the $0.5$ expectile plotted against the aridity index, elevation and the area. In all plots outliers are not included for a better visualization.

more comparable. Many of our small catchments are located in a karstic environment or moors, which exhibit highly irregular regimes that are in particular hard to regionalize. Larger catchments, in contrast, have a more regular regime and show a high prediction performance.

## 5 Conclusions

In this paper we analyzed the performance of a single spatio-temporal XGBoost model on the prediction of monthly low flow for a comprehensive dataset of 260 gauging stations in Austria. We paid particular attention to the estimation of low extremes, by applying the expectile loss function as a fitting criterion. Our results show that the expectile loss yields a high prediction accuracy, but the performance decreases strongly for small expectiles. The best performing model is the $0.5$ expectile with a $R^2_{med}$ of $0.67$, but also the $0.2$ and $0.3$ expectile reach a higher $R^2_{med}$ than the mean and median absolute loss. Small expectiles as $0.01$ or $0.025$ already yield a negative $R^2_{med}$, resulting in a high number of poor-performing stations.

Weak-performing stations can also be found for the $0.5$ expectile, where $26$ % of the stations have an $R^2$ below $0.5$. A decomposition of the model error revealed that the the monthly error is the main error component for large expectiles, while the seasonal and annual components are negligible for most stations. Considering the weak-performing stations of the $0.5$ expectile the dominant error component is a systematic error, or bias. With a median value of $56$ %, the bias is much larger at these stations compared to well-performing stations with a median value of only $11$ %. This underlines the finding that the



main shortcoming of our approach is a systematic error, which impairs the predictive accuracy. However, the strength of our spatio-temporal model is the good approximation of the seasonal and annual variability of monthly low flows, as shown by the low seasonal and annual error component.

Despite the low global performance of small expectiles ($\tau = 0.05, 0.025, 0.01$), they demonstrate an increasing accuracy when focusing on low extremes. This improvement in predictive performance has the simultaneous disadvantage that extreme low flows are increasingly misclassified. We found that the application of the expectile loss results in a trade-off between global performance, prediction accuracy of extremes, and misclassification rate of extreme events. The implementation of the $0.1$ or $0.2$ expectiles, depending on whether we want to optimize predictions for annual or more extreme low flow, appears to be optimal with respect to all criteria. The resulting extreme gradient tree boosting model covers seasonal and annual variability nicely and provides a viable approach for spatio-temporal modelling of a range of hydrological variables representing average conditions and extreme events.

We demonstrate that the expectile loss is a suitable alternative to common loss functions in spatio-temporal low flow models. However, its application is not limited to statistical learning models such as XGBoost, but can also be considered for hydrological models when their focus is on predicting hydrological extremes. As with all models, there is a trade-off between overall predictive performance, accuracy on the tails of the distribution, and identification of extreme events that should be considered in model applications.

*Code and data availability.* Data and code can be made available on personal request to johannes.laimighofer@boku.ac.at.



**Appendix A**



**Figure A1.** Predictive power of static topological predictors obtained by the backward variable selection procedure (Sect. 2.2.3). Shown are the performance gains of the 10 prediction folds per expectile model, with colors indicating whether the variable is used for the final prediction of the fold or not. Variables that were never selected by our model are not shown.

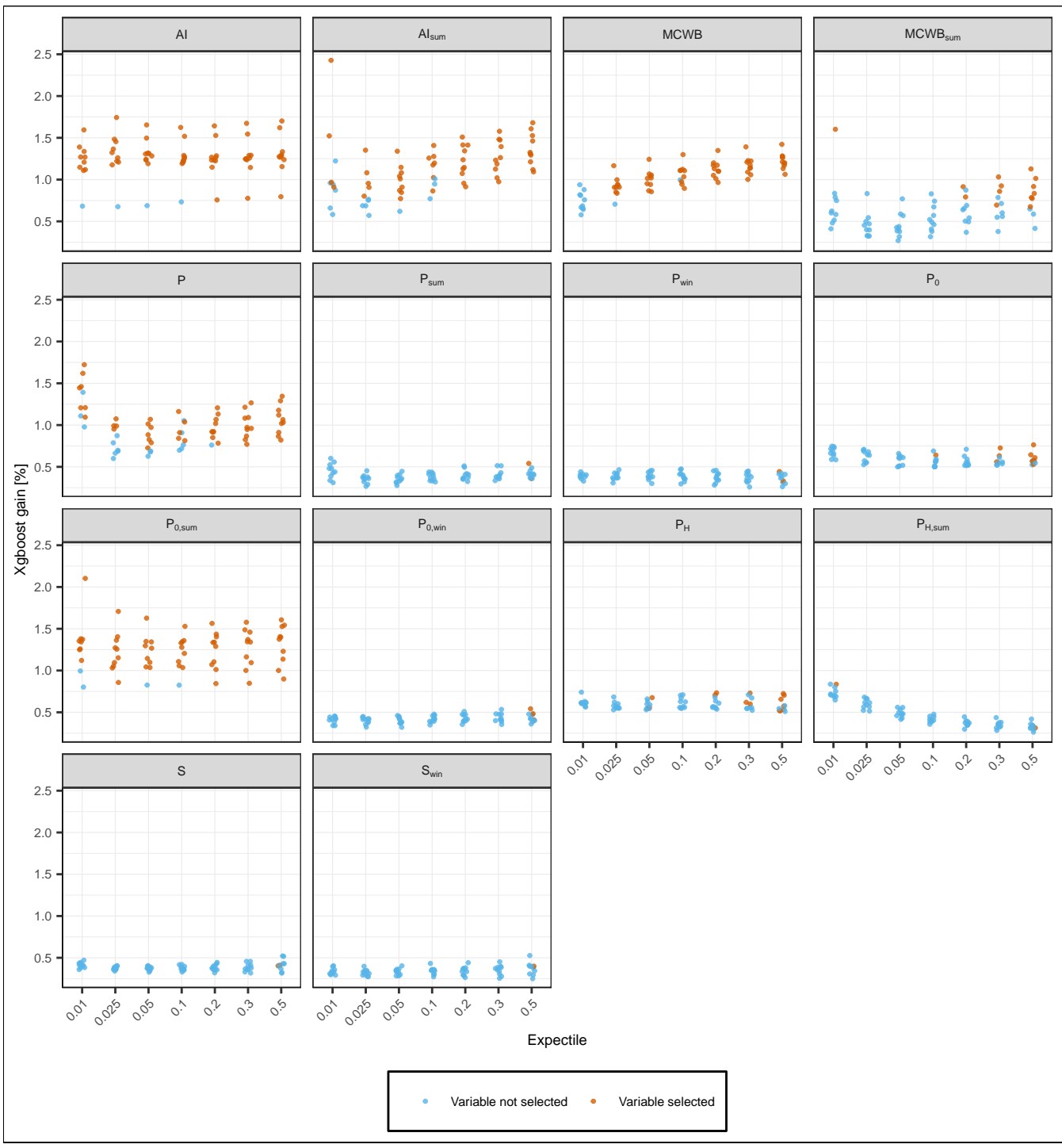

**Figure A2.** Predictive power of static meteorological predictors obtained by the backward variable selection procedure (Sect. 2.2.3). Shown are the performance gains of the 10 prediction folds per expectile model, with colors indicating whether the variable is used for the final prediction of the fold or not. Variables that were never selected by our model are not shown.





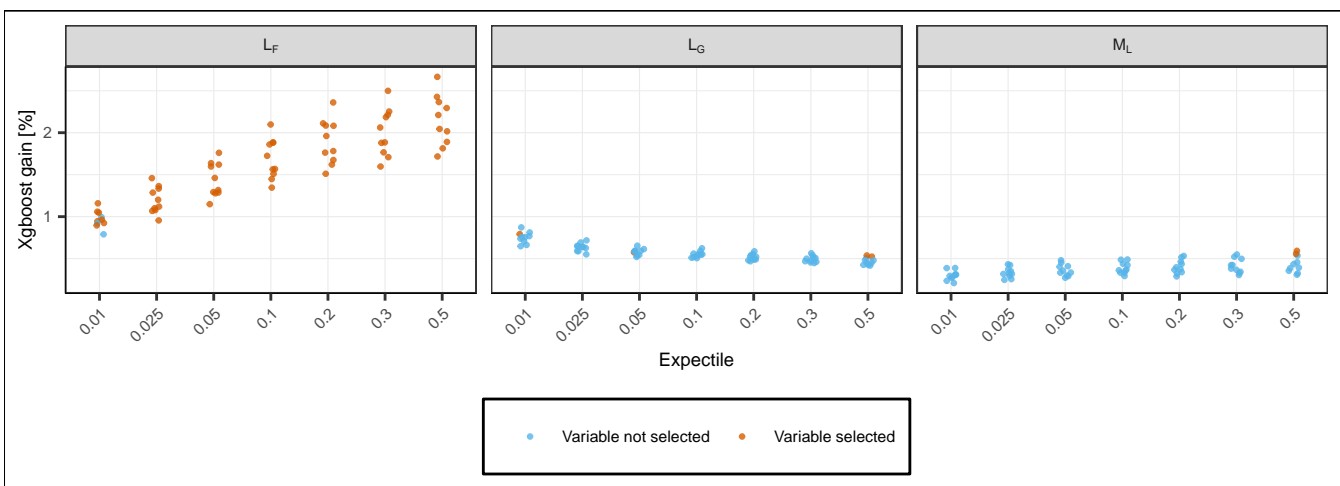

**Figure A3.** Predictive power of static landuse predictors obtained by the backward variable selection procedure (Sect. 2.2.3). Shown are the performance gains of the 10 prediction folds per expectile model, with colors indicating whether the variable is used for the final prediction of the fold or not. Variables that were never selected by our model are not shown.



*Author contributions.* JL designed the research layout and GL contributed to its conceptualization. JL performed the formal analyses, and JL and GL prepared the draft paper. MM supported the analyses. GL supervised the overall study. All the authors contributed to the interpretation of the results and writing of the paper.

*Competing interests.* We declare that we have no competing interests.

*Acknowledgements.* Johannes Laimighofer is a recipient of a DOC fellowship (grant number 25819) of the Austrian Academy of Sciences,
which is gratefully thanked for financial support.

Data provision by the Central Institute for Meteorology and Geodynamics (ZAMG) and the Hydrographical Service of Austria (HZB) was highly appreciated. This research supports the work of the UNESCO-IHP VIII FRIEND-Water program.



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
