# Peer review of "Low flow estimation beyond the mean - expectile loss and extreme gradient boosting for spatio-temporal low flow prediction in Austria"

_Hydrology and Earth System Sciences, 2022_

## Author Response (AR1)

Dear Dr. Toth,

we thank you and the referees for the constructive feedback! We updated our manuscript and uploaded a revised version accordingly. We did rewrite Section 2.1.2 including your concerns and the remarks raised by Reviewer#3.

One main point that was raised by all reviewers was the issue of collinearity. This problem was tackled by using (i) a nested cross validation scheme, where only variables are added that improve the prediction performance and the final prediction is made on an independent test set. Further, (ii) boosting algorithm have an inherent variable selection as an ensemble of weak learners is combined. Finally, (iii) the approach was tested in our previous study (Laimighofer et al. 2022).

We added clarficiations on line 138, 150, 156 and 196 and in our response to Reviewer#1 and Reviewer#2. Additionally, in our answer to Reviewer#3, on page 4 we clarified the selection of the 5 % error rule that was used in our model.

We hope our changes will make the manuscript more clear.

Best regards,
Johannes Laimighofer (on behalf of all authors)

Dear Reviewer#1,

We want to thank for this valuable and positive feedback.

*„My only concern is with the model fitting. The authors observed middling-to-high performance across a range of methods, and they also note that the methods require something like 20-30 variables for prediction. I would be interested in seeing some discussion on how the large quantity of variables may or may not be indicative of overfitting. It seems, from my arm-chair analysis here, that overfitting could explain the rapid loss in performance for extreme low flows. I'd be interested to hear what the authors have to say."*

We are aware of potential overfitting of machine learning approaches. This is why we implemented a nested cross validation procedure to reduce the possibility of overfitting the model. In our paper about parsimonious models for low flow estimation (Laimighofer et al. 2022), we showed that tree models use more variables than variable selection methods as GLM-boosting or Lasso.

It is inherent to tree models that variable selection is performed at each split individually, whereas GLM variabe selection is performed globally, for the entire model. This systematically yields a higher number of predictors in tree-based models than in linear models. From this perspective, we can safely argue that the models are not overfitted (as assured by our nested CV scheme) and the large quantity of variables may not be indicative of overfitting.

To clarify this we changed the manuscript as follows:

On line 138 we added:

The method is beneficial when predictors are collinear and is robust to overfitting.

On line 150 we added:

The shrinkage parameter $\eta$ is set to 0.1, where a small value of $\eta$ minimizes the risk of overfitting and the possibility of finding only local minima.

On line 156 we added:

Finally, we introduced an early stopping rule for the number of boosting iterations (K), where the algorithm stops when the error does not decrease for 50 iterations.

On line 196 we added:

Our method ensures that predictors are selected only when they increase the predictive performance of the model. In addition, the outer loop evaluates the predictive performance at ungauged sites independently from model fitting.

Dear Reviewer#2,

We want to thank for this valuable and positive feedback.

*My only concern is the potential existence of collinearity of temporal predictors as the authors considered CWB, CWB_center, and CWB_SDI all together as potential predictors for finding the best model. It would be good to see an analysis about if including collinear predictors yields a significant increase in the model performance compared to when only not collinear predictors are considered for the model fitting.*

Thank you for this comment. Concerning our preselection of temporal predictors, we were aware that not all temporal variables would be necessary in the sense that they will improve the predictive performance of the model, in the light of collinearity. However, the XGBoost model (Chen et al. 2016), and boosting in general (Friedmann 2001, Hastie et al. 2009, especially when used in a nested CV-approach) is known to handle collinearity of predictors through regularization parameters in a highly sophisticated way. In our approach, the inner CV loop further assures that predictors are only selected if they increase the predictive performance of the model, and the outer loop, additionally, evaluates the predictive performance at ungauged sites independently from model fitting. We have further checked that only using the CWB and the different lags yield similar performance to the presented model. For these reasons we can safely argue that collinearity is explicitly handled in our approach.

To clarify this, we changed the manuscript as follows:

On line 138 we added:

The method is beneficial when predictors are collinear and is robust to overfitting.

On line 150 we added:

The shrinkage parameter η is set to 0.1, where a small value of η minimizes the risk of overfitting and the possibility of finding only local minima.

On line 156 we added:

Finally, we introduced an early stopping rule for the number of boosting iterations (K), where the algorithm stops when the error does not decrease for 50 iterations.

On line 196 we added:

Our method ensures that predictors are selected only when they increase the predictive performance of the model. In addition, the outer loop evaluates the predictive performance at ungauged sites independently from model fitting.

*Figure 2 is not described o presented in the main text.*

This was a typo in our manuscript. The figure is now correctly referenced.

*L151: Define CV. I understand that CV means cross-validation, but that could not be evident for general readers.*

*L151: A brief description of the 10-fold CV could be helpful for the reader.*

We changed line 158f from:

The final XGBoost model was optimized in a 10-fold CV by using all parameter combinations and tuning the number of boosting iterations (number of trees).

To:

The final XGBoost model was optimized in a 10-fold cross validation (CV) by using all parameter combinations and tuning the number of boosting iterations (number of trees). A detailed description of our validation scheme is given in Sect. 2.2.3.

References

Chen, T. and Guestrin, C.: XGBoost: A Scalable Tree Boosting System, KDD '16, p. 785–794, Association for Computing Machinery, New York, NY, USA, https://doi.org/10.1145/2939672.2939785, 2016

Friedman, Jerome H. "Greedy Function Approximation: A Gradient Boosting Machine." *The Annals of Statistics* 29, no. 5 (2001): 1189–1232. http://www.jstor.org/stable/2699986.

Hastie, T., Tibshirani, R., and Friedman, J. (Eds.): The elements of statistical learning, vol. 2, Springer series in statistics New York, Springer, New York, https://doi.org/10.1007/978-0-387-84858-7, 2009.

Dear Reviewer#3,

We want to thank for this valuable and positive feedback.

*Response to SPECIFIC COMMENTS*

*L.40 p.2 – "…need of less data…"*
*I respectfully disagree about this, since data-driven models always requires a large amount of data (longer periods observations for the response variable compared to other approaches.*

We agree, the phrase was removed from the manuscript.

*L.76 p.3 … These two sentences are rather unclear to me. How this information is relevant to this study? Why refer to previous studies here? In my opinion, what is important here is to define the dataset and the preliminary analyses considered for the present study.*

We removed these two sentences from the manuscript.

*This is actually my major concern: the use of symbols and acronyms could be greatly improved throughout the manuscript, and, in truth, I found Sect. 2.1.2 rather confused. I would suggest carefully reviewing the manuscript, renaming some symbols, defining ALL the adopted symbols at their first appearance in the manuscript, providing description of some indexes if needed, and homogenizing all, avoiding the use of different symbols for the same thing.*

We have updated the manuscript, by adding explanation of the acronyms at first sight and homogenized all symbols to a more coherent structure.

*At L.103-104 p.5. Pwin and Psum should be defined; the acronym MCBW and the various subscripts should be explained (what does the "M" mean? what do "in" and "um" mean?).*

The M in MCWB stands for Mean climatic water balance. Here, we have to distinguish to the spatio-temporal covariate CWB, which is on a monthly basis and not the average. PwIN and PsUM are typos that have been changed. We additionally clarified this by adding the following on line 114f:

Note that the CWB enters the model as a static variable (MCWB) and as a spatio-temporal covariate on a monthly basis (CWB). The first serves as an intercept in the model, the second as a temporal signal that determines the monthly low flow series.

*In the caption of table 1, the subscripts "win" and "sum" are explained (they should be defined also in the text at their first appearance), but annual partitioning into two seasons has never been discussed. "Summer" is not properly the summer, but it seems to be a six months period whose start and ending times are unknown (they should be specified). The same for the "winter". Why not use the terms "dry" and "wet"? In my opinion they are better suited to the bi-seasonal division of the year.*

We added a short explanation for the two seasons. A distinction into dry and wet may be misleading in term of low flow in Austria, where low flow can  either occur in the summer half-year (April to

September) or in the winter half-year (October to March). Hence, we like to keep the distinction between summer and winter. We changed the paragraph from line 101f:

For the static predictors we used a set of climate and catchment characteristics of precedent rationalisation studies in Austria (e.g. Laaha and Blöschl, 2006; Laimighofer et al., 2022). They consist of topological and landuse variables, geological classes, and long-term average meteorological characteristics such as precipitation (P), climatic water balance (MCWB), and others (Table 1). All these variables are aggregated on an annual basis (no subscript: e.g. P, MCWB), for the summer half-year from April to September (e.g. Psum, MCWBsum), and for the winter half-year from October to March (e.g. Pwin, MCWPwin).

*In table 1, the description of some variable should be expanded. For instance SSL, SMO and SST: these three classes of slope should be defined. Have you considered some kind of slope threshold value or something else to classify the slope? Please, specify.*

We updated Table 1 and added the slope specification.

*At L.109 p.6, the symbol for the monthly climatic water balance becomes CWB. Is this different from MCWB? If not, you should homogenize the use of symbols.*

CWB referes to the monthly spatio-temporal covariate, where MCWB is the averaged climatic water balance at each station. On line 114f we added the following for an additional explanation:

Note that the CWB enters the model as a static variable (MCWB) and as a spatio-temporal covariate on a monthly basis (CWB). The first serves as an intercept in the model, the second as a temporal signal that determines the monthly low flow series.

*L.107-111 p.5-6. Here a preliminary assessment is mentioned. Please indicate how do you perform this preliminary assessment and the metric adopted to evaluate the performances.*

The preliminary assessment is based on the same validation procedure and error metrics as presented in the paper. A comparison of these different variable inputs was not scope of this paper, thereof we just added this as a side note in the methods section. As at this point of the manuscript we did not include any error metrics, nor described the validation procedure of our study, we added the following sentence on line 110:

All these combinations were tested by a nested 10-fold cross validation (CV, see Sect. 2.2.3 for more detail) and compared by a range of error metrics (Sect. 2.2.4).

*L.119 p.6. The standardized drought index should be defined. How is it computed?*

We added the computation on line 123:

Instead of fitting a parametric distribution, we estimate the empirical probability of the $CWB_{s,m}$. The empirical probabilities are then transformed to quantiles of a standard normal distribution.

*L.121 p.6. "Fig.1" should be Fig.2.*

This was a typo in our manuscript, the Figure is now correctly referenced.

*How were the range of variation of the hyperparameters (e.g. maximum depth, etc...reported between L.146 and L.149, p.8), selected? Do they come from literature? Are they default or typical ranges? What else? Please, specify.*

The ranges of the hyperparameters are mainly based on our experience with the dataset and the xgboost package. The selected range of the hyperparameters were not very sensitive to overall model performance. Using (only) values outside this range could have led to a decrease in model performance. In case of maximum depth, which can be described as a parameter for modelling interactions - a value of $1 - 3$ would be insufficient. To clarify the selected range we added the following on line 157:

The range of hyperparameters were set by preliminary experience with the XGBoost model and our individual dataset.

*L.151. (p.8). Please, specify what you exactly mean with 10-fold CV. What does the symbol CV mean? Cross-Validation? Please, add explanation to the manuscript.*

We changed line 158 from:

The final XGBoost model was optimized in a 10-fold CV by using all parameter combinations and tuning the number of boosting iterations (number of trees).

To:

The final XGBoost model was optimized in a 10-fold cross validation (CV) by using all parameter combinations and tuning the number of boosting iterations (number of trees). A detailed description of our validation scheme is given in Sect. 2.2.3.

*L.167 p.8. Double "of"*

One „of" removed.

*L.185 p.9. Are "10 CV" and "10-fold CV" the same thing? If so, please homogenize the symbols, otherwise, explain the differences.*

These two terms are the same. We changed every occurrence to 10-fold CV throughout the manuscript.

*L.85-186 p.9. How was the threshold of 1.05 selected? Please, specify.*

Model selection, hyperparameter tuning or variable selection in statistical learning/ machine learning is often based on using the minimum error. Another approach is to use a one standard error rule (e.g. Hastie et al. 2009, the standard error is added to the minimum error and the most parsimonious model with an error not higher than the minimum error + 1 standard error is selected), which yields models with less variables (in case of variable selection). Our nested CV procedure would lead to almost no reduction in the number of variables if we use the minimum error (but yielding similar performance). Using the one standard error rule yields a very low amount of variables, which impairs the prediction performance significantly. This is due the high variation of each individual fold in our dataset. Thereof, we introduced this 5 % error threshold, which basically allows the model to decrease the prediction performance, but yields model that have a lower amount of variables. This was done in a study already published in HESS (Laimighofer et al. 2022). We added the following on line 195:

This approach yields models with fewer variables at only a small loss in predictive accuracy.

*L.190 p.10. LMDAE was already explained in Sect. 2.2.2. Why introducing another symbol (MDAE) for the same thing? The same for LMAE. Consistently with the symbology adopted in the present study, "RMSE" in eq.5 should be LRMSE, and the coefficient of determination in eq. 6 should be LR2.*

We are sorry for that confusion, but in early version of the manuscript we used the LMDAE/LMAE as reference to the specific loss function. As this is not present in the current manuscript, we changed LMAE to MAE, and LMDAE to MDAE.

*Here the authors focus almost entirely in the description of the results reported in Tab. 3. I would suggest trying to expand the discussion, addressing some observed behaviors.*

We highlighted the weak performance of low expectiles as also suggested in the next but one remark by adding the following sentence on line 269:

Expectiles below 0.05 show an insufficient performance on these global error metrics. For example the RMSE of the 0.01 expectile is twice as high as the RMSE of the 0.5 expectile.

*Please, quantity the "better performance" reporting the associated values of R2med or recalling Figure 5 or Table 4 in the sentence. "Portion" at L.268 should be "fraction" or "percentage". Avoid the use of nested round brackets (..(..)..).*

The sentence (Line 282) was changed to:

However, the 0.2 and 0.3 expectile still yield a higher R2med of 0.65 and 0.6, compared to the mean absolute (0.58) and median absolute loss (0.57). Further, the fraction of stations with a weak performance (R2 < 0.5) is also lower for the 0.2 and 0.3 expectile as shown in Table 3.

*Looking at figure 5, Expectile 0.01, 0.025 and 0.05 should be discarded since they provided very few (or none for Expectile 0.01) stations with acceptable values for R2. Also the other metrics in Tab 3, at global level confirms their scarce suitability. This should be better highlighted.*

We agree that the expectiles 0.01, 0.025 and 0.05 show a insufficient performance in terms of R2 or other global error metrics. As we do not limit our study to global model performance, we prefer to keep all expectiles in the figure, despite their weak performance to present a more holistic picture. To give more emphasis on the low performance metrics of small tau values, we added the following text to the manuscript:

On line 269:

Expectiles below 0.05 show an insufficient performance on these global error metrics. For example the RMSE of the 0.01 expectile is twice as high as the RMSE of the 0.5 expectile.

On line 280:

Generally, low expectiles (0.01, 0.025, 0.05) yield a high number of inadequate models, where almost 90 % of the stations obtain a R2 below 0.5 for the 0.05 expectile. In case of the 0.01 expectile no station has a R2 greater than 0.5.

*I found this analysis very interesting, and in my opinion the discussion should be expanded a little bit trying to interpret and address the results. For example, looking at Figure 6, it seems that the main difference among the various expectile models is the produced bias, while the prediction seasonal and annual errors provide about the same relative contribution to the total error. Why? Was it expected? I would suggest to discuss the main implications in considering models with different expectiles (i.e. Expectile 0.5 instead of Expectile 0.01).*

We partially expected that low expectiles produce a larger bias, as this is part of the penalization of the expectile loss function. The issue was further discussed in Sect. 3.3 (Fig. 7, 8, 9), where we report on the results for prediction of extremes, which we consider the main implications. Finally, we discussed this in more detail in Sect. 4.1 (Fig. 11).

*In my opinion also a value of R2 of 0.42 could denote inefficient models, and, indeed, values around 0 (both negative or positive) are all representative of inefficient models and should not be compared with each other. For example, I am not sure that a model with R2 of 0.1 is better than a model providing a negative value for R2, while I rather confident that both don't work.*

We agree that a R2 value of 0.42 can denote an inefficient model and a value of 0.1 as for the 0.05 expectile sure does. To make this more clear, we changed the sentence on Line 313 to:

For example if we assess the accuracy of our models for cases below the 1 % quantile, the 0.01 expectile is yielding at least a R2 of 0.42, where larger expectiles (0.05 - 0.5) show inefficient models with R2 values below 0.1.

*I would suggest to provide a possible explanation that may justify the observed increasing trend for the hit score and decreasing trend for the precision index with decreasing expectile (from 0.5 to 0.01) and equal quantile. The authors should probably consider the definitions of the two metrics (see eqs. 12 and 13), their meaning and implications in terms of models prediction ability. Looking for an optimal trade-off solution, which model should be preferred? Which metric carries the highest weight in describing models prediction ability considering the different target that the model may have?*

*In the last sentence the authors state "...the user needs to find some optimum"(L.330), but in my opinion authors should provide here some practical indications to do so. This important aspect is roughly mentioned only in the conclusion section.*

We discussed the precision score on line 380-382 and added a sentence for the Hit score on line 381:

In contrast, the hit score drops sharply for higher expectiles, resulting in a low detection rate of extreme events.

The optimal trade-off solution for our model is discussed on Line 385f. We also referenced, that the optimal model solution may depend on the aim of the specific model.

*L.340 p.19. Please, explain what you exactly mean with "...variable importance of less than 2 %...". Which metric is considered in this analysis to assess the % gain?*

The gain was shortly explained in Sect. 2.2.3 Line 189. To make this more clear we added the following explanation to Line 189:

The variables are ranked after their additive gain in minimizing the loss function over the 500 boosting steps. For a more robust approach, the initial variable ranking is averaged over 25 boostrap samples. The gain of each variable for the final variable ranking is the ratio of the individual additive gain to the total gain over all variables.

*First paragraph (from L.379 to L.393. P.22). Here, it seems that the authors compare quantitatively different metrics (i.e. coefficient of determination and Nash and Sutcliffe Efficiency). I think that the comparison can be done quantitatively only using the same metric, otherwise it can be done only qualitatively, for example considering some performance rating able to classify the different performances (e.g. "weak", "good", excellent", etc.), which usually are specific for the variable under consideration, the adopted metric and the time and space scale of analysis.*

We agree that if the coefficient of determination is defined by the linear relationship between predictions and observations, it only would be possible to compare the results qualitatively. Further, using the linear relationship between predictions and observations would have led to larger R2 values, the bias would not be included in our performance metric, and we would have no values below 0. Hence, in Equ. 6, we introduced the R2 as cross-validated metric, which has the same definition as the NSE. We are sorry if this has led to a misunderstanding and to add clarification we added the following on line 204:

… and the coefficient of determination R2, which - by our definition - is the same as the Nash-Sutcliffe-Efficiency (NSE) (Blöschl et al., 2013).

*L.388 P.22. "Sicilia" should be "Sicily".*

Sicilia was changed to Sicily.

*L.418-420 P.23. Given the high number of predictor variables considered, could overfitting partially explain the overall performance drop for extreme low flow and for decreasing expectiles (e.g. 0.01 or 0.025)? Perhaps, the authors might add something about this "potential" problem and how they tried to prevent overfitting.*

This issue was already raised by Reviewer#1 and Reviewer#2. We added some explanations to the manuscript and for more detailed answers, please have a look at our replies to Reviewer#1 and Reviewer#2.

References

Hastie, T., Tibshirani, R., and Friedman, J. (Eds.): The elements of statistical learning, vol. 2, Springer series in statistics New York, Springer, New York, https://doi.org/10.1007/978-0-387-84858-7, 2009.